# Neighborhood Gradient Mean: An Efficient Decentralized Learning Method for Non-IID Data

**Sai Aparna Aketi**[*]    *saketi@purdue.edu*
*Department of Electrical and Computer Engineering*
*Purdue University*

**Sangamesh Kodge**[*]    *skodge@purdue.edu*
*Department of Electrical and Computer Engineering*
*Purdue University*

**Kaushik Roy**    *kaushik@purdue.edu*
*Department of Electrical and Computer Engineering*
*Purdue University*

**Reviewed on OpenReview:** *https://openreview.net/forum?id=vkiKzK5G3e*

## Abstract

Decentralized learning algorithms enable the training of deep learning models over large distributed datasets, without the need for a central server. The current state-of-the-art decentralized algorithms mostly assume the data distributions to be Independent and Identically Distributed (IID). In practical scenarios, the distributed datasets can have significantly different data distributions across the agents. This paper focuses on improving decentralized learning on non-IID data with minimal compute and memory overheads. We propose *Neighborhood Gradient Mean (NGM)*, a novel decentralized learning algorithm that modifies the local gradients of each agent using self- and cross-gradient information. In particular, the proposed method averages the local gradients with model-variant or data-variant cross-gradients based on the communication budget. Model-variant cross-gradients are derivatives of the received neighbors' model parameters with respect to the local dataset. Data-variant cross-gradient derivatives of the local model with respect to its neighbors' datasets. The data-variant cross-gradients are aggregated through an additional communication round. We theoretically analyze the convergence characteristics of *NGM* and demonstrate its efficiency on non-IID data sampled from various vision and language datasets. Our experiments demonstrate that the proposed method either remains competitive or outperforms (by $0-6\%$) the existing state-of-the-art (SoTA) decentralized learning algorithm on non-IID data with significantly less compute and memory requirements. Further, we show that the model-variant cross-gradient information available locally at each agent can improve the performance on non-IID data by $2-20\%$ without additional communication cost.

## 1 Introduction

The remarkable success of deep learning is mainly attributed to the availability of humongous amounts of data and computing power. Large amounts of data are generated on a daily basis at different devices all over the world which could be used to train powerful deep learning models. Collecting such data for centralized processing is not practical because of communication and privacy constraints. To address this concern, a new interest in developing distributed learning algorithms (Agarwal & Duchi, 2011) has emerged. Federated learning (centralized learning) (Konecny et al., 2016) is a popular setting in the distributed machine learning

---

[*]These authors contributed equally to this work

paradigm, where the training data is kept locally at the edge devices and a global shared model is learned by aggregating the locally computed updates through a coordinating central server. Such a setup requires continuous communication with a central server which becomes a potential bottleneck (Haghighat et al., 2020). This has motivated the advancements in decentralized machine learning.

Decentralized machine learning is a branch of distributed learning that focuses on learning from data distributed across multiple agents/devices. Unlike Federated learning, these algorithms assume that the agents are connected peer to peer without a central server. It has been demonstrated that decentralized learning algorithms (Lian et al., 2017) can perform comparably to centralized algorithms on benchmark vision datasets. Lian et al. (2017) present Decentralised Parallel Stochastic Gradient Descent (D-PSGD) by combining SGD with gossip averaging algorithm (Xiao & Boyd, 2004). Further, the authors analytically show that the convergence rate of D-PSGD is similar to its centralized counterpart (Dean et al., 2012). Decentralized Momentum Stochastic Gradient Descent (DMSGD) which introduces momentum to D-PSGD was proposed and analyzed in Balu et al. (2021). Assran et al. (2019) introduce Stochastic Gradient Push (SGP) which extends D-PSGD to directed and time-varying graphs. Tang et al. (2019); Koloskova et al. (2019) explore error-compensated compression techniques (Deep-Squeeze and CHOCO-SGD) to reduce the communication cost of D-PSGD significantly while achieving the same convergence rate as centralized algorithms. Aketi et al. (2021) combined Deep-Squeeze with SGP to propose communication-efficient decentralized learning over time-varying and directed graphs. Lu & De Sa (2020); Liu et al. (2020b); Zhao et al. (2022); Takezawa et al. (2023) also explore communication compression in decentralized setups. Recently, Koloskova et al. (2020) proposed a unified framework for the analysis of gossip-based decentralized SGD methods and provide the best-known convergence guarantees.

The key assumption to achieve state-of-the-art performance by all the above-mentioned decentralized algorithms is that the data is independent and identically distributed (IID) across the agents. In particular, the data is assumed to be distributed in a uniform and random manner across the agents. This assumption does not hold in most of the real-world applications as the data distributions across the agents are significantly different (non-IID) based on the user pool (Hsieh et al., 2020). The effect of non-IID data in a peer-to-peer decentralized setup is a relatively under-studied problem. There are only a few works that try to bridge the performance gap between IID and non-IID data for a decentralized setup. Note that, we mainly focus on a common type of non-IID data, widely used in prior works (Tang et al., 2018; Lin et al., 2021; Esfandiari et al., 2021): a skewed distribution of data labels across agents. Tang et al. (2018) proposed $D^2$ algorithm that extends D-PSGD to non-IID data. However, the algorithm was demonstrated on only a basic LeNet5 (LeCun et al., 1998) model and is not scalable to deeper models with normalization layers. SwarmSGD proposed by Nadiradze et al. (2019) leverages random interactions between participating agents in a graph to achieve consensus. Lin et al. (2021) replace local momentum with Quasi-Global Momentum (QGM) and improve the test performance by $1-20\%$. However, the improvement in accuracy is only $1-2\%$ in case of highly skewed data distribution as shown in Aketi et al. (2022). Tracking mechanisms such as Gradient Tracking (Di Lorenzo & Scutari, 2016; Pu & Nedić, 2021; Koloskova et al., 2021) and Momentum Tracking (Takezawa et al., 2022) have been proposed to tackle non-IID data in decentralized settings at the cost of $2\times$ communication overhead. Most recently, Esfandiari et al. (2021) proposed Cross-Gradient Aggregation (CGA) and a compressed version of CGA (CompCGA), claiming state-of-the-art performance for decentralized learning algorithms over completely non-IID data. CGA aggregates *cross-gradient* information, i.e., derivatives of its model with respect to its neighbors' datasets through an additional communication round. It then updates the model using projected gradients based on quadratic programming. CGA and CompCGA require a very slow quadratic programming step (Goldfarb & Idnani, 1983) after every iteration for gradient projection which is both compute and memory intensive. This work focuses on the following question: *Can we improve the performance of decentralized learning on non-IID data with minimal compute and memory overhead?*

In this paper, we propose *Neighborhood Gradient Mean* (*NGM*) algorithm with two variants to handle non-IID data in peer-to-peer decentralized learning setups. Firstly, we classify the gradients at each agent into three types, namely self-gradients, model-variant cross-gradients, and data-variant cross-gradients (see Section 3). The self-gradients (or local gradients) are the derivatives computed at each agent on its model parameters with respect to the local dataset. The model-variant cross-gradients are the derivatives of the

received neighbors' model parameters with respect to the local dataset. These gradients are computed locally at each agent after receiving the neighbors' model parameters. Communicating the neighbors' model parameters is a necessary step in any gossip-based decentralized algorithm (Lian et al., 2017). The data-variant cross-gradients are the derivatives of the local model with respect to its neighbors' datasets. These gradients are obtained through an additional round of communication. We then cluster the gradients into a) *model-variant cluster* with self-gradients and model-variant cross-gradients, and b) *data-variant cluster* with self-gradients and data-variant cross-gradients. Finally, the local gradients are replaced with the model-variant cluster means in $NGM_{mv}$ where the communication budget is $1\times$, and with the data-variant cluster means in $NGM_{dv}$ where the communication budget is $2\times$. The main motivation behind this modification is to account for the high variation in the computed local gradients (and in turn the model parameters) across the neighbors due to the non-IID nature of the data distribution.

The proposed $NGM_{dv}$ has two rounds of communication at every iteration to send model parameters and data-variant cross-gradients. This incurs $2\times$ communication cost compared to traditional decentralized algorithms (D-PSGD). To reduce this communication overhead, we show that $NGM_{dv}$ can be combined with error feedback-based compression methods to compress the additional round of cross-gradient communication. Note that $NGM_{mv}$ has no communication overhead. We provide a detailed convergence analysis of the proposed $NGM_{dv}$ algorithm and show that the analysis can be trivially extended to $NGM_{mv}$. We validate the performance of the proposed algorithm on various datasets, model architectures, and graph topologies. We compare the proposed algorithm with several baselines such as D-PSGD (Lian et al., 2017), QG-DSGDm (Lin et al., 2021), Momentum Tracking (Takezawa et al., 2022), and CGA (Esfandiari et al., 2021). Our experiments show that $NGM$ can achieve superior performance on non-IID data compared to the current state-of-the-art approaches. We also report the order of communication, memory, and compute overheads required for various decentralized algorithms as compared to D-PSGD.

**Contributions:** In summary, we make the following contributions.

- We propose the Neighborhood Gradient Mean ($NGM$) algorithm with two variations for decentralized learning on non-IID data. $NGM_{mv}$ utilizes self-gradients and model-variant cross-gradients to improve the performance on non-IID data. $NGM_{dv}$ improves this performance further by utilizing self-gradients and data-variant cross-gradients at the cost of $2\times$ communication overhead.

- We theoretically show that the convergence rate of $NGM$ is $\mathcal{O}(\frac{1}{\sqrt{NK}})$, which is consistent with the state-of-the-art decentralized learning algorithms.

- $NGM_{mv}$ outperforms the QG-DSGDm baseline by $2 - 20\%$ without any communication overhead.

- $NGM_{dv}$ either remains competitive or outperforms by $0 - 6\%$ with significantly less compute and memory requirements compared to the current state-of-the-art at iso-communication.

## 2 Background

In this section, we provide the background on decentralized learning algorithms with peer-to-peer connections.

The main goal of decentralized machine learning is to learn a global model using the knowledge extracted from the locally generated and stored data samples across $N$ edge devices/agents while maintaining privacy constraints. In particular, we solve the optimization problem of minimizing global loss function $\mathcal{F}(x)$ distributed across $N$ agents as given in equation. 1.

$$\min_{x \in \mathbb{R}^d} \mathcal{F}(x) = \frac{1}{N} \sum_{i=1}^{N} f_i(x),$$
$$and \ \ f_i(x) = \mathbb{E}_{d^i \in D^i}[F_i(x; d^i)] \ \ \forall i \tag{1}$$

This is typically achieved by combining stochastic gradient descent (Bottou, 2010) with global consensus-based gossip averaging (Xiao & Boyd, 2004). The communication topology in this setup is modeled as a

graph $G = ([N], E)$ with edges $\{i, j\} \in E$ if and only if agents $i$ and $j$ are connected by a communication link exchanging the messages directly. We represent $\mathcal{N}_i$ as the neighbors of $i$ including itself. It is assumed that the graph $G$ is strongly connected with self-loops i.e., there is a path from every agent to every other agent. The adjacency matrix of the graph $G$ is referred to as a mixing matrix $W$ where $w_{ij}$ is the weight associated with the edge $\{i, j\}$. Note that, weight 0 indicates the absence of a direct edge between the agents. We assume that the mixing matrix is doubly-stochastic and symmetric, similar to all previous works in decentralized learning. For example, in a undirected ring topology, $w_{ij} = \frac{1}{3}$ if $j \in \{i - 1, i, i + 1\}$. Further, the initial models and all the hyperparameters are synchronized at the beginning of the training. Algorithm. 2 in the appendix describes the flow of D-PSGD with momentum. The convergence of the Algorithm. 2 assumes the data distribution across the agents to be Independent and Identically Distributed (IID).

## 3    Neighborhood Gradient Mean

We propose the *Neighborhood Gradient Mean (NGM)* algorithm and a compressed version of *NGM* which improve the performance of decentralized learning on non-IID data. We define the concepts of self-gradients and cross-gradients below.

**Self-Gradient:** For an agent $i$ with the local dataset $D^i$ and model parameters $x^i$, the self-gradient is the gradient of the loss function $f_i$ with respect to the model parameters $x^i$, evaluated on mini-batch $d^i$ sampled from dataset $D^i$.

$$g^{ii} = \nabla_x F_i(x^i; d^i) \tag{2}$$

**Cross-Gradient:** For an agent $i$ with model parameters $x^i$ connected to neighbor $j$ that has local dataset $D^j$, the cross-gradient is the gradient of the loss function $f_j$ with respect to the model parameters $x^i$, evaluated on mini-batch $d^j$ sampled from dataset $D^j$.

$$g^{ij} = \nabla_x F_j(x^i; d^j) \tag{3}$$

At each agent $i$, we further divide the cross-gradients into two categories. 1) *Model-variant cross-gradients*: The derivatives that are computed locally using its local data on the neighbors' model parameters ($g^{ji}$). 2) *Data-variant cross-gradients*: The derivatives (received through communication) of its model parameters on the neighbors' dataset ($g^{ij}$). Note that an agent $i$ computes the cross-gradients $g^{ji}$ that act as model-variant cross-gradients for $i$ and as data-variant cross-gradients for $j$. In particular, the data-variant cross-gradient of $i$ i.e., $g^{ij}$ is computed by agent $j$ using its local data $d_j$ after receiving the model parameters $x^i$ from $i$ and is then communicated to agent $i$ if needed.

### 3.1    The *NGM* algorithm

The pseudo-code of the Neighborhood Gradient Mean (*NGM*) is shown in Algorithm. 1. The overview of the algorithm is illustrated in Figure. 5 of Appendix A.3.

The main contribution of the proposed *NGM* algorithm is the local gradient manipulation step (line 12 in Algorithm. 1). We first introduce a hyper-parameter $\alpha$ called *NGM mixing weight* which is either set to 0 or 1. In the $k^{th}$ iteration of *NGM*, each agent $i$ calculates its self-gradient $g^{ii}$. Then, agent $i$'s model parameters are transmitted to all other agents ($j$) in its neighborhood, and the respective cross-gradients are calculated by the neighbors. These cross-gradients are transmitted back to agent $i$ through an additional communication round if $\alpha$ is set to 1. At every iteration after the first communication round, each agent $i$ has access to self-gradients ($g^{ii}$) and model-variant cross-gradients. When $\alpha$ is set to 1, the agents get access to data-variant cross-gradients through a second round of communication. Now, we group these cross-gradients into two clusters: a) *Model-variant cluster* $\{g^{ji} \forall j \in \mathcal{N}_i\}$ that includes self-gradients and model-variant cross-gradients, and b) *Data-variant cluster* $\{g^{ij} \forall j \in \mathcal{N}_i\}$ that includes self-gradients and data-variant cross-gradients. The local gradients at each agent are replaced with either of the clusters' mean based on $\alpha$ value as shown in Equation. 4, which assumes a uniform mixing matrix ($w_{ij} = 1/m; m = |\mathcal{N}_i|$).

---

**Algorithm 1** Neighborhood Gradient Mean (*NGM*)

---

**Input:** Each agent $i \in [1, N]$ initializes model weights $x^i_{(0)}$, step size $\eta$, momentum coefficient $\beta$, averaging rate $\gamma$, mixing matrix $W = [w_{ij}]_{i,j\in[1,N]}$, *NGM* mixing weight $\alpha \in \{0, 1\}$, and $I_{ij}$ are elements of $N \times N$ identity matrix, $\mathcal{N}_i$ represents neighbors of $i$ including itself.

Each agent simultaneously implements the TRAIN( ) procedure
1. **procedure** TRAIN( )
2.     **for** k=0, 1, ..., $K - 1$ **do**
3.         $d^i_k \sim D^i$
4.         $g^{ii}_k = \nabla_x f_i(d^i_k; x^i_k)$
5.         SENDRECEIVE($x^i_k$)
6.         **for** each neighbor $j \in \{\mathcal{N}_i - i\}$ **do**
7.             $g^{ji}_k = \nabla_x f_i(d^i_k; x^j_k)$
8.             **if** $\alpha \neq 0$: SENDRECEIVE($g^{ji}_k$)
11.         **end**
12.         $\widetilde{g}^i_k = \sum_{j\in\mathcal{N}_i}[(1-\alpha)w_{ji}g^{ji}_k + \alpha w_{ij}g^{ij}_k]$
13.         $v^i_k = \beta v^i_{(k-1)} - \eta\widetilde{g}^i_k$
14.         $\widetilde{x}^i_k = x^i_k + v^i_k$
15.         $x^i_{(k+1)} = \widetilde{x}^i_k + \gamma\sum_{j\in\mathcal{N}_i}(w_{ij} - I_{ij})x^j_k$
16.     **end**
17. **return**

---

When $\alpha$ is set to zero, the proposed algorithm modifies the local gradient with model-variant cluster mean and this is referred to as $NGM_{mv}$. This version of the algorithm does not incur any communication overhead The $\alpha = 1$ version of the algorithm replaces local gradient with data-variant cluster mean and is referred to as $NGM_{dv}$. Setting $\alpha$ to any non-zero value requires an additional round of communication and we experimentally determine that $\alpha = 1$ gives the best results (refer section. 6 for more details).

$$\widetilde{g}^i_k = (1-\alpha)\underbrace{\left[\frac{1}{m}\sum_{j\in\mathcal{N}_i} g^{ji}_k\right]}_{\text{(a) Model-variant cluster mean}} + \alpha\underbrace{\left[\frac{1}{m}\sum_{j\in\mathcal{N}_i} g^{ij}_k\right]}_{\text{(b) Data-variant cluster mean}} \tag{4}$$

The motivation for this modification is to reduce the variation of the computed local gradients across the agents. In IID settings, the local gradients should statistically resemble the cross-gradients and hence simple gossip averaging is sufficient to reach convergence. However, in the non-IID case, the local gradients across the agents are significantly different due to the variation in datasets and hence the model parameters on which the gradients are computed. The proposed algorithm reduces this variation in the local gradients as it is equivalent to adding the bias term $\epsilon$ for $NGM_{mv}$ and the bias term $\omega$ for $NGM_{dv}$ as shown in Equation. 5.

$$\widetilde{g}^i_k = g^{ii}_k + (1-\alpha)\underbrace{\left[\frac{1}{m}\sum_{j\in\mathcal{N}_i}(g^{ji}_k - g^{ii}_k)\right]}_{\text{model variance bias }\epsilon^i_k} + \alpha\underbrace{\left[\frac{1}{m}\sum_{j\in\mathcal{N}_i}(g^{ij}_k - g^{ii}_k)\right]}_{\text{data variance bias }\omega^i_k}$$

$$\epsilon^i_k = \frac{1}{m}\sum_{j\in\mathcal{N}_i}\left(\nabla_x F(d^j_k; x^j_k) - \nabla_x F(d^i_k; x^i_k)\right) \tag{5}$$

$$\omega^i_k = \frac{1}{m}\sum_{j\in\mathcal{N}_i}\left(\nabla_x F(d^j_k; x^i_k) - \nabla_x F(d^i_k; x^i_k)\right)$$

The bias term $\epsilon$ compensates for the difference in a neighborhood's self-gradients caused due to variation in the model parameters across the neighbors. Whereas, the bias term $\omega$ compensates for the difference in a neighborhood's self-gradients caused due to variation in the data distribution across the neighbors. We hypothesize and show through our experiments that the addition of one of these bias terms to the local gradients improves the performance of decentralized learning on non-IID data by accelerating global convergence.

## 4 Convergence Analysis of *NGM*

In this section, we provide a convergence analysis for $NGM_{dv}$. We assume that the following statements hold:

**Assumption 1 - Lipschitz Gradients:** Each function $f_i(x)$ is L-smooth.

**Assumption 2 - Bounded Variance:** The variance of the stochastic gradients is assumed to be bounded.

$$\mathbb{E}_{d \sim D_i} ||\nabla F_i(x; d) - \nabla f_i(x)||^2 \leq \sigma^2 \;\; \forall i \in [1, N] \tag{6}$$

$$||\nabla f_i(x) - \nabla \mathcal{F}(x)||^2 \leq \zeta^2 \;\; \forall i \in [1, N] \tag{7}$$

**Assumption 3 - Doubly Stochastic Mixing Matrix:** The mixing matrix $W$ is a real doubly stochastic matrix with $\lambda_1(W) = 1$ and

$$max\{|\lambda_2(W)|, |\lambda_N(W)|\} \leq \sqrt{\rho} < 1 \tag{8}$$

where $\lambda_i(W)$ is the $i^{th}$ largest eigenvalue of W and $\rho$ is a constant.

The above assumptions are commonly used in most decentralized learning setups.

**Lemma 1.** *Given assumptions 1-3, we define $g^i = \nabla F_i(x; d)$ and the NGM gradient update $\widetilde{g}^i$. For all $i$, we have:*

$$\mathbb{E}\left[\left\|\frac{1}{N} \sum_{i=1}^{N} (\tilde{\mathbf{g}}^i - \mathbf{g}^i)\right\|^2\right] \leq 4(\frac{\sigma^2}{N} + \zeta^2) \tag{9}$$

A complete proof for Lemma 1 can be found in Appendix. A.1. This lemma bounds the difference between self-gradients and the proposed gradients in terms of inter- and intra-gradient variance bounds. The inter-gradient variation bounded by $\zeta^2$ determines the heterogeneity in the data distribution i.e., the non-IIDness. The intra-gradient variation bounded by $\sigma^2$ determines the stochastic noise across the mini-batches within an agent. Intuitively, the distance between the self-gradients $g^i$ and the $NGM_{dv}$ gradient update $\widetilde{g}^i$ increases with an increase in the degree of heterogeneity in data distribution which is expected. Note that the convergence rate for $NGM_{mv}$ remains the same as CGA where $\mathbb{E}\left[\left\|(\tilde{\mathbf{g}}^i - \mathbf{g}^i)\right\|^2\right]$ is bounded by a positive constant which is of the order of $\frac{1}{K}$.

Theorem 1 presents the convergence of the proposed $NGM_{dv}$ algorithm and the proof is detailed in Appendix A.2.2

**Theorem 1.** *(Convergence of $NGM_{dv}$ algorithm) Given assumptions 1-3 and lemma 1, let step size $\eta$ satisfy the following conditions:*

$$(1) \;\; \eta \leq \frac{\sqrt{4(1 - \sqrt{\rho})^2 + 6(1 - \sqrt{\rho})(1 - \beta)^2} - 2(1 - \sqrt{\rho})}{6L}$$

$$(2) \;\; 1 - \frac{L\eta}{2}(\sqrt{1 + \frac{4}{L\eta}} - 1) < \beta < 1$$

*For all $K \geq 1$, we have*

$$\frac{1}{K} \sum_{k=0}^{K-1} \mathbb{E}[||\nabla \mathcal{F}(\bar{x}_k)||^2] \leq \frac{1}{C_1 K} (\mathcal{F}(\bar{x}^0) - \mathcal{F}^*) + C_2(\frac{\sigma^2}{N} + \zeta^2) + C_3 \frac{(\sigma^2 + \zeta^2)}{(1 - \sqrt{\rho})^2} \tag{10}$$

*where $\beta = momentum\ coefficient$, $\bar{x}_k = \frac{1}{N} \sum_{i=1}^{N} x_k^i$*

$C_1 = \frac{1}{2}\left(\frac{\eta}{1-\beta} - \frac{(1-\beta)}{\beta L}\right)$, $C_2 = \left(\frac{10\eta^2 L(\eta^2+\beta)}{(1-\beta)^3}\right)/C_1 = \frac{20\eta^2 L^2 \beta(\eta^2+\beta)}{(1-\beta)^2(\eta\beta L-(1-\beta)^2)}$, and $C_3 = \frac{14\eta^3 L^3 \beta}{(1-\beta)^2(\eta\beta L-(1-\beta)^2)}$.

The result of the theorem. 1 shows that the magnitude of the average gradient achieved by the consensus model is upper-bounded by the difference between the initial objective function value and the optimal value, the sampling variance, and gradient variations across the agents representing data heterogeneity. A detailed explanation of the constraints on step size and momentum coefficient is presented in the Appendix. A.2.3. Further, we present a corollary to show the convergence rate of $NGM_{dv}$ in terms of the number of iterations.

**Corollary 1.** *Suppose that the step size satisfies* $\eta = \mathcal{O}\left(\sqrt{\frac{N}{K}}\right)$ *and that* $\zeta^2 = \mathcal{O}\left(\frac{1}{(1-\sqrt{\rho})N}\right)$. *For a sufficiently large* $K \geq \frac{36NL^2}{r^2}$, *and*
$r = \sqrt{4(1-\sqrt{\rho})^2 + 6(1-\sqrt{\rho})(1-\beta)^2} - 2(1-\sqrt{\rho})$, *we have, for some constant* $C > 0$,

$$\frac{1}{K}\sum_{k=0}^{K-1}\mathbb{E}\left[\|\nabla \mathcal{F}(\bar{\mathbf{x}}_k)\|^2\right] \leq C\left(\frac{1}{\sqrt{NK}} + \frac{1}{K}\right).$$

The proof for Corollary. 1 can be found in Appendix. A.2.4. Here, the constant $C$ depended on the spectral gap $\rho$, stochastic noise bound $\sigma^2$ and the inter gradient variance bound $\zeta^2$. The bound on $\zeta^2$ is the result of the assumption that the graphs with higher connectivity (lower spectral gap) can tolerate more non-IIDness. Corollary. 1 indicates that the proposed algorithm achieves linear speedup (with a convergence rate of $\mathcal{O}(\frac{1}{\sqrt{NK}})$) when $K$ is sufficiently large. This convergence rate is similar to the well-known best result for decentralized SGD algorithms in the literature as shown in Table. 1.

Table 1: Convergence rate comparison between decentralized learning algorithms.

| Method | Rate | Assumption | Simplified Rate |
|---|---|---|---|
| D-PSGD | $\mathcal{O}\left(\frac{1+\sigma^2}{\sqrt{NK}} + \frac{N^2\sigma^2}{K(1-\rho)} + \frac{N^2\zeta^2}{K(1-\sqrt{\rho})^2}\right)$ | - | $\mathcal{O}(\frac{1}{\sqrt{NK}} + \frac{1}{K})$ |
| CGA | $\mathcal{O}\left(\frac{1+\sigma^2}{\sqrt{NK}} + \frac{N(\zeta^2+\sigma^2+\epsilon^2)}{K(1-\sqrt{\rho})^2} + (\frac{\sqrt{K}}{\sqrt{N}} + \frac{\sqrt{N}}{\sqrt{K}})\epsilon^2\right)$ | $\epsilon^2 = \mathcal{O}(\frac{1}{K})$ | $\mathcal{O}(\frac{1}{\sqrt{NK}} + \frac{1}{K} + \frac{1}{K^{1.5}} + \frac{1}{K^2})$ |
| $NGM_{dv}$ (ours) | $\mathcal{O}\left(\frac{1+\sigma^2}{\sqrt{NK}} + \frac{N(\sigma^2+\zeta^2)}{K(1-\sqrt{\rho})^2} + \frac{\sqrt{N}}{\sqrt{K}}\zeta^2\right)$ | $\zeta^2 = \mathcal{O}\left(\frac{1}{(1-\sqrt{\rho})N}\right)$ | $\mathcal{O}(\frac{1}{\sqrt{NK}} + \frac{1}{K})$ |

## 5 Experiments

In this section, we analyze the performance of the proposed $NGM_{mv}$, and $NGM_{dv}$ techniques and compare them with the corresponding baseline i.e., D-PSGD algorithm (Lian et al., 2017), QG-DSGDm (Lin et al., 2021), Momentum Tracking (Takezawa et al., 2022) and state-of-the-art CGA method (Esfandiari et al., 2021). [*]

### 5.1 Experimental Setup

The efficiency of the proposed method is demonstrated through our experiments on a diverse set of datasets, model architectures, tasks, topologies, and numbers of agents. We present the analysis on – (a) Datasets (Appendix A.5): vision datasets (CIFAR-10, CIFAR-100, Fashion MNIST and Imagenette (Husain, 2018)) and language datasets (AGNews (Zhang et al., 2015)). (b) Model architectures (Appendix A.6): 5-layer CNN, VGG-11, ResNet-20, LeNet-5, MobileNet-V2, ResNet-18, BERT$_{mini}$ and DistilBERT$_{base}$ (c) Tasks: Image and Text Classification. (d) Topologies: Ring, Chain, and Torus. (e) Number of agents: varying from 4 to 20. Note that we use low-resolution ($32 \times 32$) images of the Imagenette dataset for the experiments in Table. 3. The results for high resolution ($224 \times 224$) Imagenette are presented in Table. 4.

---

[*]Our PyTorch code is available at `https://github.com/aparna-aketi/neighborhood_gradient_clustering`

We consider an extreme case of non-IID distribution where no two neighboring agents have the same class. This is referred to as complete label-wise skew or 100% label-wise non-IIDness (Hsieh et al., 2020). In particular, for a 10-class dataset such as CIFAR-10 - each agent in a 5-agent system has data from 2 distinct classes, and each agent in a 10-agent system has data from a unique class. For a 20-agent system, two agents that are maximally apart share the samples belonging to a class. We report the test accuracy of the consensus model averaged over three randomly chosen seeds. The details of the hyperparameters for all the experiments are present in Appendix. A.10. We compare the proposed method with iso-communication baselines. The experiments on $NGM_{mv}$ are compared with D-PSGD and QG-DSGDm, $NGM_{dv}$ with Momentum Tracking and CGA. The communication cost for each experiment in this section is presented in Appendix A.9.

## 5.2 Results

Firstly, we evaluate variants of $NGM$ and compare them with respective baselines in Table. 2, for training different models trained on CIFAR-10 over various graph sizes and topologies. We observe that $NGM_{mv}$ consistently outperforms QG-DSGDm for all models and graph sizes over ring topology with a significant performance gain varying from $2 - 20\%$. For the torus topology, $NGM_{mv}$ outperforms D-PSGD by $14 - 18\%$ and QG-DSGDm fails to converge. Our experiments show the superiority of $NGM_{dv}$ over CGA. The performance gains are more pronounced when considering larger graphs (with 20 agents) and compact models such as ResNet-20. We observe that CGA and $NGM$ perform better than Momentum Tracking in most cases. We further demonstrate the generalizability of the proposed method by evaluating it on various image datasets (refer Table. 3) such as Fashion MNIST, and Imagenette and on challenging datasets such as CIFAR-100. Table. 3, 4 show that $NGM_{mv}$ outperforms QG-DSGDm by $1 - 18\%$ across various datasets whereas $NGM_{dv}$ remain competitive with an average improvement of $\sim 1\%$.

Table 2: Average test accuracy comparisons for CIFAR-10 with non-IID data using various architectures and graph topologies. The results are averaged over three seeds where std is indicated.

| Method | Agents | 5layer CNN Ring | 5layer CNN Torus | VGG-11 Ring | ResNet-20 Ring |
|---|---|---|---|---|---|
| D-PSGD | 5 | $76.00 \pm 1.44$ | - | $67.04 \pm 5.36$ | $82.13 \pm 0.84$ |
| | 10 | $47.68 \pm 3.20$ | $55.34 \pm 6.32$ | $44.14 \pm 3.30$ | $31.66 \pm 6.01$ |
| | 20 | $44.85 \pm 1.94$ | $50.12 \pm 1.91$ | $38.92 \pm 2.99$ | $31.94 \pm 2.91$ |
| QG-DSGDm | 5 | $78.77 \pm 0.69$ | - | $81.71 \pm 0.45$ | $82.59 \pm 2.66$ |
| | 10 | $47.83 \pm 3.39$ | $14.48 \pm 0.50$ | $57.77 \pm 0.33$ | $46.09 \pm 7.90$ |
| | 20 | $46.57 \pm 4.80$ | $17.25 \pm 5.81$ | $32.32 \pm 7.61$ | $44.78 \pm 3.39$ |
| $NGM_{mv}$ (ours) | 5 | $\mathbf{82.20 \pm 0.34}$ | - | $\mathbf{83.65 \pm 0.38}$ | $\mathbf{85.88 \pm 0.58}$ |
| | 10 | $\mathbf{67.43 \pm 1.15}$ | $\mathbf{73.84 \pm 0.33}$ | $\mathbf{59.92 \pm 2.12}$ | $\mathbf{66.02 \pm 2.86}$ |
| | 20 | $\mathbf{58.80 \pm 1.30}$ | $\mathbf{64.55 \pm 1.16}$ | $\mathbf{52.70 \pm 1.63}$ | $\mathbf{50.74 \pm 2.36}$ |
| Momentum Tracking | 5 | $80.34 \pm 0.67$ | - | $79.62 \pm 7.42$ | $85.65 \pm 0.30$ |
| | 10 | $58.54 \pm 7.51$ | $67.46 \pm 1.10$ | $73.51 \pm 1.42$ | $66.37 \pm 7.72$ |
| | 20 | $36.58 \pm 11.5$ | $56.71 \pm 1.96$ | $36.12 \pm 13.8$ | $44.35 \pm 11.7$ |
| CGA | 5 | $82.20 \pm 0.43$ | - | $84.41 \pm 0.22$ | $87.52 \pm 0.50$ |
| | 10 | $72.96 \pm 0.40$ | $76.04 \pm 0.62$ | $\mathbf{79.66 \pm 0.46}$ | $79.98 \pm 1.23$ |
| | 20 | $69.88 \pm 0.84$ | $73.21 \pm 0.27$ | $79.30 \pm 0.12$ | $75.13 \pm 1.56$ |
| $NGM_{dv}$ (ours) | 5 | $\mathbf{83.36 \pm 0.65}$ | - | $\mathbf{85.15 \pm 0.58}$ | $\mathbf{88.52 \pm 0.19}$ |
| | 10 | $\mathbf{75.34 \pm 0.30}$ | $\mathbf{78.53 \pm 0.56}$ | $79.55 \pm 0.30$ | $\mathbf{84.02 \pm 0.44}$ |
| | 20 | $\mathbf{73.36 \pm 0.88}$ | $\mathbf{75.11 \pm 0.07}$ | $\mathbf{79.43 \pm 0.62}$ | $\mathbf{81.26 \pm 0.69}$ |

To show the effectiveness of the proposed method across different modalities, we present results on the text classification task in Table 4. We train on the $BERT_{mini}$ model with the AGNews dataset distributed over 4 and 8 agents and $DistilBert_{base}$ distributed over 4 agents. For $NGM_{mv}$, we see a maximum improvement of 2.1% over the baseline D-PSGD algorithm. Even for the text classification task, we observe $NGM_{dv}$ to be competitive with CGA. These observations are consistent with the results of the image classification

Table 3: Average test accuracy comparisons for various datasets with non-IID sampling trained over undirected ring topology. The results are averaged over three seeds where std is indicated.

| Method | Agents | Fashion MNIST (LeNet-5) | CIFAR-100 (ResNet-20) | Imagenette (32×32) (MobileNet-V2) |
|---|---|---|---|---|
| D-PSGD | 5 | $86.43 \pm 0.14$ | $44.66 \pm 5.23$ | $47.09 \pm 9.20$ |
| | 10 | $75.49 \pm 0.32$ | $19.03 \pm 13.27$ | $32.81 \pm 2.18$ |
| QG-DSGDm | 5 | $89.28 \pm 0.52$ | $49.60 \pm 3.30$ | $43.70 \pm 2.80$ |
| | 10 | $84.38 \pm 0.58$ | $41.28 \pm 0.61$ | $17.80 \pm 2.66$ |
| $NGM_{mv}$ (ours) | 5 | $\mathbf{90.03 \pm 0.11}$ | $\mathbf{55.96 \pm 0.95}$ | $\mathbf{60.15 \pm 2.17}$ |
| | 10 | $\mathbf{87.10 \pm 0.47}$ | $\mathbf{49.30 \pm 1.02}$ | $\mathbf{36.13 \pm 1.97}$ |
| Momentum Tracking | 5 | $90.52 \pm 0.56$ | $49.52 \pm 2.36$ | $38.79 \pm 2.21$ |
| | 10 | $\mathbf{87.82 \pm 0.19}$ | $37.74 \pm 4.74$ | $23.07 \pm 11.3$ |
| CGA | 5 | $90.03 \pm 0.39$ | $56.43 \pm 2.39$ | $72.82 \pm 1.25$ |
| | 10 | $87.61 \pm 0.30$ | $53.61 \pm 1.07$ | $61.97 \pm 0.58$ |
| $NGM_{dv}$ (ours) | 5 | $\mathbf{90.61 \pm 0.18}$ | $\mathbf{56.50 \pm 3.23}$ | $\mathbf{74.49 \pm 0.93}$ |
| | 10 | $87.24 \pm 0.23$ | $\mathbf{53.77 \pm 0.15}$ | $\mathbf{64.06 \pm 1.11}$ |

tasks. Finally, through this exhaustive set of experiments, we demonstrate that the $NGM_{dv}$ can serve as a simple plugin to boost the performance of decentralized learning on non-IID data without significant memory and compute overheads. Further, $NGM_{mv}$ demonstrates that locally available model-variant cross-gradient information at each agent can be efficiently utilized to improve decentralized learning with no communication overhead.

Table 4: Average test accuracy comparisons for AGNews dataset (left side of the table) and full resolution ($224 \times 224$) Imagenette dataset (right side of the table). The results are averaged over three seeds where std is indicated.

| Method | AGNews-BERT$_{mini}$ Agents = 4 Ring Topology | AGNews-BERT$_{mini}$ Agents = 8 Ring Topology | AGNews-DistilBERT$_{base}$ Agents = 4 Ring Topology | Imagenette (224×224)-ResNet-18 Agents = 5 Ring Topology | Imagenette (224×224)-ResNet-18 Agents = 5 Chain Topology |
|---|---|---|---|---|---|
| D-PSGD | $89.21 \pm 0.41$ | $85.48 \pm 0.71$ | $91.54 \pm 0.07$ | $65.43 \pm 4.60$ | $42.02 \pm 1.25$ |
| QG-DSGDm | $87.38 \pm 5.29$ | $76.61 \pm 3.63$ | $93.55 \pm 0.23$ | $74.22 \pm 2.30$ | $43.35 \pm 4.57$ |
| $NGM_{mv}$ (ours) | $\mathbf{89.40 \pm 0.13}$ | $\mathbf{87.58 \pm 0.07}$ | $\mathbf{93.60 \pm 0.12}$ | $\mathbf{78.26 \pm 0.67}$ | $\mathbf{47.87 \pm 0.99}$ |
| Momentum Tracking | $82.45 \pm 6.19$ | $87.82 \pm 0.047$ | $93.87 \pm 0.15$ | $81.11 \pm 0.29$ | $22.12 \pm 2.93$ |
| CGA | $91.43 \pm 0.11$ | $\mathbf{89.15 \pm 0.45}$ | $93.42 \pm 0.04$ | $85.00 \pm 0.67$ | $65.96 \pm 1.84$ |
| $NGM_{dv}$ (ours) | $\mathbf{92.24 \pm 0.29}$ | $89.02 \pm 0.39$ | $\mathbf{94.11 \pm 0.01}$ | $\mathbf{85.85 \pm 0.60}$ | $\mathbf{67.77 \pm 1.76}$ |

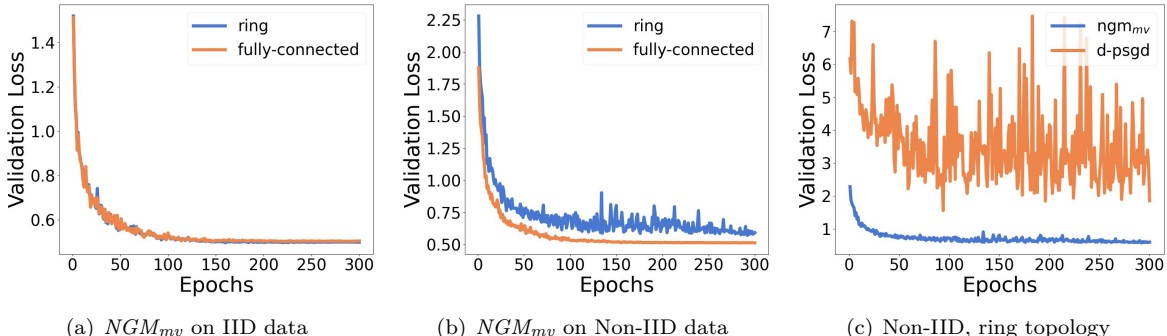

(a) $NGM_{mv}$ on IID data

(b) $NGM_{mv}$ on Non-IID data

(c) Non-IID, ring topology

Figure 1: Average Validation loss of $NGM_{mv}$ during training CIFAR-10 on a 5-layer CNN with 5 agents.

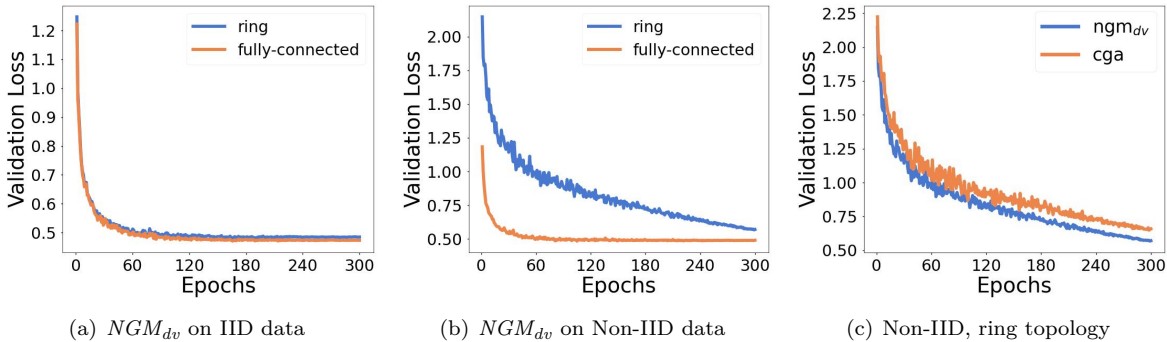

(a) $NGM_{dv}$ on IID data     (b) $NGM_{dv}$ on Non-IID data     (c) Non-IID, ring topology

Figure 2: Average Validation loss of $NGM_{dv}$ during training CIFAR-10 on a 5-layer CNN with 5 agents.

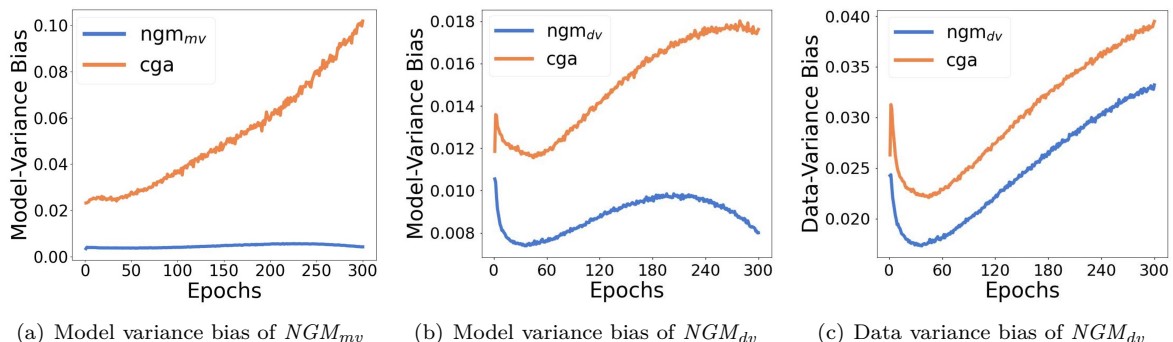

(a) Model variance bias of $NGM_{mv}$     (b) Model variance bias of $NGM_{dv}$     (c) Data variance bias of $NGM_{dv}$

Figure 3: Ablation study: CIFAR-10 dataset trained on a 5-layer CNN over ring graph 5 agents.

### 5.3 Analysis

We empirically show the generalization characteristics in terms of validation loss convergence for the proposed algorithm on IID and Non-IID data in Figures. 1(a), 2(a), and Figures. 1(b), 2(b) respectively. For Non-IID data, we observe that there is a slight difference in convergence (as expected) with a slower rate for sparser topology (ring graph) compared to a denser counterpart (fully connected graph). Figure. 1(c), 2(c) shows the comparison of the generalization characteristics of the $NGM_{mv}$ and $NGM_{dv}$ algorithm with the respective baselines. For the same decentralized setup, We observe that $NGM_{mv}$ and $NGM_{dv}$ have lower validation loss than D-PSGD and CGA respectively. Analysis for 10 agents is presented in Appendix A.7.

Then we proceed to analyze the model variance and data variance bias terms when training with various decentralized methods. Figure. 3(a) shows that the model variance bias of $NGM_{mv}$ is much lower than of the D-PSGD baseline resulting in the better performance of $NGM_{mv}$. We then compare the model variance and data variance bias terms of $NGM_{dv}$ with CGA as shown in Figure. 3(b), and 3(c) respectively. We observe that both the model variance and the data variance bias for $NGM_{dv}$ are significantly lower than CGA. This is because CGA gives more importance to self-gradients as it updates in the descent direction that is close to self-gradients and is positively correlated to data-variant cross-gradients. In contrast, $NGM$ accounts for the biases directly and gives equal importance to self-gradients and data-variant cross-gradients, thereby achieving superior performance. Further, Figure.4(a) presents the average consensus error i.e., $\frac{1}{n}||x_k^i - \bar{x}_k||_F^2$ over time for training CIFAR-10 dataset on 5-layer CNN with respect to various algorithms. This empirically shows that the proposed $NGM$ algorithm reaches the consensus at the same rate as CGA.

Additionally, we show that $NGM$ can be used in synergy with QGM (Lin et al., 2021) to further improve the performance of decentralized learning on non-IID Data. Figure. 4(b) shows the test accuracy on the CIFAR-10 dataset trained on ResNet-20 over ring topology. We observe that the QGM version of $NGM_{mv}$ performs better than $NGM_{mv}$ by $16 - 27\%$. Similarly, the QGM version of $NGM_{dv}$ performs $1.3 - 2.5\%$ better than the local momentum version of $NGM_{dv}$.

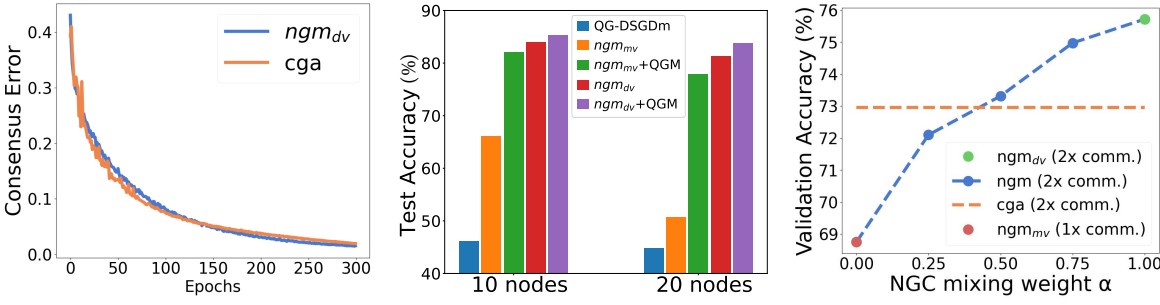

(a) Consensus Error with 5-layer CNN over 5 agents

(b) Training ResNet-20 by combining *NGM* with Quasi Global Momentum

(c) Ablation study on $\alpha$ with 5-layer CNN over 5 agents

Figure 4: Training CIFAR-10 distributed in a non-IID fashion on a ring topology

## 5.4 Communication Compression

In this section, we discuss the impact of communication compression on the proposed *NGM* algorithm. At every iteration, the $NGM_{dv}$ algorithm involves two rounds of communication with the neighbors: 1) communicate the model parameters, and 2) communicate the cross-gradients. This communication overhead can be a bottleneck in a resource-constrained environment. Hence, we explore a compressed version of *NGM* referred to as *CompNGM* using Error Feedback SGD (EF-SGD) (Karimireddy et al., 2019) to compress the cross-gradients. *CompNGM* compresses the error-compensated cross-gradients from 32 bits (floating point precision of arithmetic operations) to 1 bit by using scaled signed gradients. The pseudo-code for *CompNGM* is shown in Algorithm. 3 in the Appendix. Additional results and analysis on *CompNGM* are presented in the Appendix. A.4.

Table 5: Test Accuracy of *CompNGM*, Accuracy drop compared to $NGM_{dv}$ and communication reduction in Giga Bytes compared to $NGM_{dv}$ for various datasets trained on ring topology.

| Dataset | Model | 5 agents | | | 10 agents | | |
|---------|-------|----------|--|--|-----------|--|--|
| | | Acc (%) | Acc. Drop (%) | Comm. Drop (GB) | Acc (%) | Acc. Drop (%) | Comm. Drop (GB) |
| Fashion MNIST | LeNet-5 | $90.48 \pm 0.19$ | 0.13 | 16.71 (1.94×) | $83.38 \pm 0.39$ | 3.86 | 8.35 (1.94×) |
| CIFAR-10 | ResNet-20 | $87.56 \pm 0.34$ | 0.96 | 123.11 (1.94×) | $78.50 \pm 0.98$ | 5.52 | 61.75 (1.94×) |
| CIFAR-100 | ResNet-20 | $57.51 \pm 0.48$ | -1.01 | 100.49 (1.94×) | $43.07 \pm 0.32$ | 10.7 | 50.27 (1.94×) |
| Imagenette (32×32) | MobileNet-V2 | $72.91 \pm 1.06$ | 1.58 | 99.89 (1.94×) | $61.91 \pm 2.10$ | 2.15 | 49.98 (1.94×) |

Table. 5 shows that *CompNGM* reduces the communication cost by 1.94× by trading off $0.1 - 1.57\%$ test accuracy for 5 agents and $2-10\%$ test accuracy for 10 agents. This shows that *NGM* can be combined with communication compression techniques to achieve a trade-off between accuracy boost and communication overhead. Further, *NGM* can be combined with stronger compression methods such as EF21 (Richtárik et al., 2021) to have a better trade-off between accuracy and communication cost.

## 5.5 Hardware Benefits

The proposed $NGM_{dv}$ algorithm is superior in terms of memory and compute efficiency (see Table. 6 and Appendix. A.8) while having equal communication cost as compared to *CGA*. $NGM_{dv}$ replaces the complicated quadratic projection step of CGA with a simple weighted averaging step. Since $NGM_{dv}$ involves weighted averaging, an additional buffer to store the cross-gradients is not required. CGA stores all the cross-gradients in a matrix form for quadratic programming projection of the local gradient. Therefore, *NGM* has no memory overhead compared to the baseline D-PSGD algorithm, while CGA requires additional memory equivalent to the number of neighbors times model size. Moreover, the quadratic programming projection step (Goldfarb & Idnani, 1983) in CGA is much more expensive in compute and latency than the weighted averaging step of cross-gradients in *NGM*. Our evaluation clearly shows that *NGM* is superior to CGA in terms of test accuracy, memory efficiency, compute efficiency, and latency.

Table 6: Comparison of communication, memory, and compute overheads per mini-batch compared to D-PSGD. $m_s$: model size, $n_i$: number of neighbors, $b$: floating point precision, Q: compute for Quadratic Programming, B: compute for Backward Pass. Note that $\mathcal{O}(Q) >> \mathcal{O}(n_i m_s)$ (Nesterov & Todd, 1997)

| Method | Comm. | Memory | Compute |
|---|---|---|---|
| QG-DSGDm | 0 | $\mathcal{O}(m_s)$ | $\mathcal{O}(n_i m_s)$ |
| $NGM_{mv}$ | 0 | 0 | $\mathcal{O}(n_i m_s + n_i B)$ |
| Momentum Tracking | $\mathcal{O}(n_i m_s)$ | $\mathcal{O}(2m_s)$ | $\mathcal{O}(n_i m_s)$ |
| CGA | $\mathcal{O}(n_i m_s)$ | $\mathcal{O}(n_i m_s)$ | $\mathcal{O}(Q + n_i B)$ |
| $NGM_{dv}$ | $\mathcal{O}(n_i m_s)$ | 0 | $\mathcal{O}(n_i m_s + n_i B)$ |
| CompCGA | $\mathcal{O}(\frac{n_i m_s}{b})$ | $\mathcal{O}(n_i m_s)$ | $\mathcal{O}(Q + n_i B)$ |
| $CompNGM$ | $\mathcal{O}(\frac{n_i m_s}{b})$ | $\mathcal{O}(n_i m_s)$ | $\mathcal{O}(n_i m_s + n_i B)$ |

## 6  Discussion and Limitations

As mentioned in Section. 3.1, the $NGM$ mixing weight $\alpha$ is either set to 0 or 1 resulting in $NGM_{mv}$ or $NGM_{dv}$ respectively. This implies that $NGM_{mv}$ utilizes model-variant cross-gradients whereas, $NGM_{dv}$ utilizes data-variant cross-gradients. However, the $NGM_{dv}$ variant of the algorithm has access to both model-variant and data-variant cross-gradients. So, we conduct an ablation study mixing these two cross gradients by varying $NGM$ mixing weight $\alpha$ in the range of $[0, 1]$. Figure. 4(c) depicts the change in average validation accuracy with $\alpha$ and we observe that $\alpha = 1$ performs best. Further, Figure. 3(b) shows that the model variant bias reduces for $NGM_{dv}$ even though it does not explicitly utilize the model-variant cross-gradients. We hypothesize that, in $NGM_{dv}$, the use of data-variant cross-gradients for local updates followed by gossip averaging of model parameters inherently adds model-variant cross-gradients reducing model variance. Hence, explicitly mixing the two sets of cross-gradients is not necessary. In summary, for $1\times$ communication constraints i.e., when agents do not have access to data-variant cross-gradients, $NGM_{mv}$ effectively utilizes locally available model-variant cross-gradients to improve the accuracy. On the contrary, when $2\times$ communication is allowed (non-zero $\alpha$), the best performance is achieved by only utilizing the data-variant cross-gradients.

Table 7: Test Accuracy and Latency of ImageNet (non-IID) trained on ResNet-18 over 10-agent Ring. Note, LM: Local Momentum, QGM: Quasi Global Momentum.

| Trained for 60 epochs | D-PSGD | QG-DSGDm | $NGM_{mv}$ | $NGM_{mv}$ +LM | $NGM_{mv}$ + QGM | Momentum Tracking | CGA | $NGM_{dv}$ + LM | $NGM_{dv}$ + QGM |
|---|---|---|---|---|---|---|---|---|---|
| Accuracy (%) | 25.02 | 45.19 | 42.66 | 49.22 | 51.00 | 43.65 | - | 49.42 | 54.31 |
| Epoch Time (min) | 7.6 | 7.9 | 18.6 | 18.8 | 19.1 | 10.7 | 406 | 21.5 | 22.2 |

$NGM_{mv}$ has one potential limitation compared to D-PSGD and QG-DSGDm i.e., compute overhead. $NGM_{mv}$ has to compute model-variant cross-gradients which requires every agent $i$ to compute $n_i$ additional forward and backward passes. However, $NGM_{dv}$ is significantly better in terms of latency, compute, and memory requirements compared to CGA as it simplifies the complicated quadratic projection step. Running the CGA algorithm on larger datasets like ImageNet is impractical due to the significant latency associated with the quadratic projection step. Table. 7 reports the accuracy and latency of training ImageNet in a non-IID fashion over a ring topology of 10 agents. We observe that CGA is $\sim 20\times$ slower than $NGM_{dv}$ for the ImageNet dataset on ResNet-18. We estimate the training time for one simulation of CGA on ImageNet (90 epochs with 10-agent ring topology) to be around 26 days on 4 Nvidia A100s (80 GB) with Intel(R) Xeon(R) Gold 6338 CPU. Further, it has been shown in the literature (Koloskova et al., 2019; Tang et al., 2019) that the communication compression with error feedback only affects the higher-order terms of the convergence rate. Hence, decentralized compression algorithms have linear speedup similar to D-PSGD. Following this, our future efforts go towards showing that $CompNGM$ has a similar convergence rate as $NGM$. We would also like to explore efficient methods to compute cross-gradients for $NGM$.

## 7 Conclusion

Enabling decentralized training on non-IID data is key for ML applications to efficiently leverage the humongous amounts of user-generated private data. In this paper, we propose the *Neighborhood Gradient Mean* (*NGM*) algorithm with two variants that improve decentralized learning on non-IID data. $NGM_{mv}$ improves the performance of decentralized learning utilizing model-variant cross-gradients without any communication overhead. $NGM_{dv}$ further increases these gains by operating on data-variant cross-gradients. Additionally, we present a compressed version of our algorithm (*CompNGM*) to reduce the communication overhead associated with data-variant cross-gradients of $NGM_{dv}$. We theoretically analyze the convergence characteristic and empirically validate the performance of the proposed techniques over different model architectures, datasets, graph sizes, and topologies. Finally, we compare the proposed algorithms with the current state-of-the-art decentralized learning algorithm on non-IID data and show superior performance with significantly less compute and memory requirements setting the new state-of-the-art for decentralized learning on non-IID data.

## Authors' Contributions:

Sai Aparna Aketi and Sangamesh Kodge, both worked on developing the algorithm. The experiments for Computer Vision tasks were designed by Sai Aparna Aketi and the experiments for Natural Language Processing tasks were designed by Sangamesh Kodge. The theory for the convergence analysis and consensus error bounds was developed by Sai Aparna Aketi and was verified by Sangamesh Kodge. All the authors contributed equally in writing and proofreading the paper.

## Acknowledgments

This work was supported in part by, the Center for Brain-inspired Computing (C-BRIC), the Center for the Co-Design of Cognitive Systems (COCOSYS), a DARPA-sponsored JUMP center, the Semiconductor Research Corporation (SRC), the National Science Foundation, and DARPA ShELL.

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

# A Appendix

## A.1 Proof of Lemma. 1

This section presents detailed proof for Lemma. 1. The *NGM* algorithm modifies the local gradients as follows

$$\widetilde{g}^i = (1-\alpha)\sum_{j\in\mathcal{N}_i} w_{ji}g^{ji} + \alpha\sum_{j\in\mathcal{N}_i} w_{ij}g^{ij} = g^i + \sum_{j\in\mathcal{N}_i} w_{ij}(g^{ij} - g^i) \quad (\text{NGM}_{\text{dv}} \text{ uses } \alpha = 1)$$

$$\widetilde{g}^i - g^i = \sum_{j\in\mathcal{N}_i} w_{ij}(\nabla F_j(x, d^j) - \nabla F_i(x, d^i))$$

Now we prove the lemma. 1 by applying expectation to the above inequality

$$\mathbb{E}\left[\left\|\frac{1}{N}\sum_{i=1}^{N}(\tilde{\mathbf{g}}^i - \mathbf{g}^i)\right\|^2\right] = \mathbb{E}\left[\left\|\frac{1}{N}\sum_{i=1}^{N}\sum_{j\in\mathcal{N}_i} w_{ij}(\nabla F_j(x, d^j) - \nabla F_i(x, d^i))\right\|^2\right]$$

$$\stackrel{(a)}{=} \mathbb{E}\left[\left\|\frac{1}{N}\sum_{i=1}^{N}\sum_{j\in\mathcal{N}_i} w_{ij}(\nabla F_j(x, d^j) - \nabla F_i(x, d^i) - \mathbb{E}[\nabla F_j(x, d^j) - \nabla F_i(x, d^i)])\right\|^2\right]$$

$$+ \mathbb{E}\left[\left\|\frac{1}{N}\sum_{i=1}^{N}\sum_{j\in\mathcal{N}_i} w_{ij}(\mathbb{E}[\nabla F_j(x, d^j) - \nabla F_i(x, d^i)])\right\|^2\right]$$

$$\stackrel{(b)}{=} \mathbb{E}\left[\left\|\frac{1}{N}\sum_{i=1}^{N}\sum_{j\in\mathcal{N}_i} w_{ij}(\nabla F_j(x, d^j) - \nabla f_j(x)) - (\nabla F_i(x, d^i) - \nabla f_i(x))\right\|^2\right]$$

$$+ \mathbb{E}\left[\left\|\frac{1}{N}\sum_{i=1}^{N}\sum_{j\in\mathcal{N}_i} w_{ij}(\nabla f_j(x) - \nabla f_i(x))\right\|^2\right]$$

$$\stackrel{(c)}{=} \frac{1}{N^2}\sum_{i=1}^{N}\mathbb{E}\left[\left\|\sum_{j\in\mathcal{N}_i} w_{ij}(\nabla F_j(x, d^j) - \nabla f_j(x)) - (\nabla F_i(x, d^i) - \nabla f_i(x))\right\|^2\right]$$

$$+ \mathbb{E}\left[\left\|\frac{1}{N}\sum_{i=1}^{N}\sum_{j\in\mathcal{N}_i} w_{ij}(\nabla f_j(x) - \nabla f_i(x))\right\|^2\right]$$

$$\stackrel{(d)}{\leq} \frac{1}{N^2}\sum_{i=1}^{N}\sum_{j\in\mathcal{N}_i} w_{ij}\mathbb{E}\left[\left\|\nabla F_j(x, d^j) - \nabla f_j(x) - (\nabla F_i(x, d^i) - \nabla f_i(x))\right\|^2\right]$$

$$+ \frac{1}{N}\sum_{i=1}^{N}\sum_{j\in\mathcal{N}_i} w_{ij}\mathbb{E}\left[\left\|\nabla f_j(x) - \nabla\mathcal{F}(x) + \nabla\mathcal{F}(x) - \nabla f_i(x)\right\|^2\right]$$

$$\stackrel{(e)}{\leq} \frac{1}{N^2}\sum_{i=1}^{N}\sum_{j\in\mathcal{N}_i} w_{ij}4\sigma^2 + \frac{1}{N}\sum_{i=1}^{N}\sum_{j\in\mathcal{N}_i} w_{ij}\mathbb{E}\left[\left\|\nabla f_j(x) - \nabla\mathcal{F}(x) + \nabla\mathcal{F}(x) - \nabla f_i(x)\right\|^2\right]$$

$$= \frac{4N\sigma^2}{N^2} + \frac{1}{N}\sum_{i=1}^{N}\sum_{j\in\mathcal{N}_i} w_{ij}\mathbb{E}\left[\left\|\nabla f_j(x) - \nabla\mathcal{F}(x) + \nabla\mathcal{F}(x) - \nabla f_i(x)\right\|^2\right] \quad (\because \sum_{j\in\mathcal{N}_i} w_{ij} = 1)$$

$$\stackrel{(f)}{\leq} \frac{4\sigma^2}{N} + \frac{1}{N}\sum_{i=1}^{N}\sum_{j\in\mathcal{N}_i} w_{ij}\left(2\mathbb{E}\left[\left\|\nabla f_j(x) - \nabla\mathcal{F}(x)\right\|^2\right] + 2\mathbb{E}\left[\left\|\nabla f_i(x) - \nabla\mathcal{F}(x)\right\|^2\right]\right)$$

$$\stackrel{(g)}{\leq} \frac{4\sigma^2}{N} + \frac{1}{N}\sum_{i=1}^{N}\sum_{j\in\mathcal{N}_i} w_{ij}4\zeta^2$$

$$= \frac{4\sigma^2}{N} + 4\zeta^2$$

(a) identity $\mathbb{E}[||Z||^2] = \mathbb{E}[||Z - E[Z]||^2] + \mathbb{E}[||\mathbb{E}[Z]||^2]$ holds for any random vector Z; (b) the fact $\mathbb{E}[\nabla F_i(x, d^i)] = \nabla f_i(x)$; (c) $\nabla F_i(x, d^i) - \nabla f_i(x)$ are independent random vectors with 0 mean and $\mathbb{E}[||\sum_{i=1}^{N} Z_i||] = \sum_{i=1}^{N} \mathbb{E}[||Z_i||^2]$ holds when $Z_i$ are independent with mean zero; (d) jensen's inequality; (e) follows from assumption 2 $\mathbb{E}_{d \sim D_i}||\nabla F_i(x; d) - \nabla f_i(x)||^2 \leq \sigma^2 \ \forall i \in [1, N]$; (f) the basic inequality $||a_1 + a_2||^2 \leq 2||a_1||^2 + 2||a_2||^2$ for any vectors $a_1$, $a_2$; and (g) follows from assumption 2 that $||\nabla f_i(x) - \nabla \mathcal{F}(x)||^2 \leq \zeta^2 \ \forall i \in [1, N]$

$\therefore$ we have following bound given by lemma. 1: $\mathbb{E}[||\widetilde{g}^i - g^i||^2] \leq 4(\frac{\sigma^2}{N} + \zeta^2)$

### A.2    Convergence Analysis

In this section, we present the proof for our main theorem. 1 indicating the convergence of the proposed $NGM_{dv}$ algorithm. Before we proceed to the proof of the theorem, we present a lemma showing that $NGM$ achieves consensus among the different agents.

#### A.2.1    Bound for Consensus Error

**Lemma 2.** *Given assumptions 1-3 and lemma 1, the distance between the average sequence iterate $\bar{x}^k$ and the sequence iterates $x_i^k$'s generated NGM (i.e., the consensus error of the proposed algorithm) is given by*

$$\sum_{k=0}^{K} \frac{1}{N} \sum_{i=1}^{N} \mathbb{E}[||\bar{x}_k - x_k^i||^2] \leq \frac{14\eta^2(\sigma^2 + \zeta^2)}{(1-\beta)^2(1-\sqrt{\rho})^2} K + \frac{6\eta^2}{(1-\beta)^2(1-\sqrt{\rho})} \sum_{k=0}^{K-1} \mathbb{E}[||\frac{1}{N} \sum_{i=1}^{N} \nabla f_i(\mathbf{x}_k^i)||^2] \quad (11)$$

*where $K \geq 1$ and $\beta \in [0, 1)$ is the momentum coefficient.*

To prove Lemma 2 and Theorem. 1, we follow the similar approach as (Esfandiari et al., 2021). Hence, we also define the following auxiliary sequence along with Lemma 3

$$\bar{\mathbf{z}}_k := \frac{1}{1-\beta}\bar{\mathbf{x}}_k - \frac{\beta}{1-\beta}\bar{\mathbf{x}}_{k-1} \quad (12)$$

Where $k > 0$ and $\mathbf{x}_k$ is obtained by multiplying the update law by $\frac{1}{N}\mathbf{1}\mathbf{1}^\top$, ($\mathbf{1}$ is the column vector with entries being 1).

$$\bar{\mathbf{v}}_k = \beta\bar{\mathbf{v}}_{k-1} - \eta\frac{1}{N} \sum_{i=1}^{N} \tilde{\mathbf{g}}_{k-1}^i$$
$$\bar{\mathbf{x}}_k = \bar{\mathbf{x}}_{k-1} + \bar{\mathbf{v}}_k \quad (13)$$

If $k = 0$ then $\bar{\mathbf{z}}^k = \bar{\mathbf{x}}^k$. For the rest of the analysis, the initial value will be directly set to 0.

To prove Lemma. 2, we use the following facts.

$$\bar{\mathbf{z}}_{k+1} - \bar{\mathbf{z}}_k = -\frac{\eta}{1-\beta}\frac{1}{N} \sum_{i=1}^{N} \tilde{\mathbf{g}}_k^i. \quad (14)$$

$$\sum_{k=0}^{K-1} ||\bar{\mathbf{z}}_k - \bar{\mathbf{x}}_k||^2 \leq \frac{\eta^2\beta^2}{(1-\beta)^4} \sum_{k=0}^{K-1} \left\|\frac{1}{N} \sum_{i=1}^{N} \tilde{\mathbf{g}}_k^i\right\|^2. \quad (15)$$

Note that the proof for Eq. 14 and Eq. 15 can be found in (Esfandiari et al., 2021) as lemma-3 and lemma-4 respectively.

Before proceeding to prove Lemma 2, we introduce some key notations and facts that serve to characterize the Lemma similar to (Esfandiari et al., 2021).

We define the following notations:

$$
\begin{aligned}
\tilde{\mathbf{G}}_k &\triangleq [\tilde{\mathbf{g}}_k^1, \tilde{\mathbf{g}}_k^2, ..., \tilde{\mathbf{g}}_k^N] \\
\mathbf{V}_k &\triangleq [\mathbf{v}_k^1, \mathbf{v}_k^2, ..., \mathbf{v}_k^N] \\
\mathbf{X}_k &\triangleq [\mathbf{x}_k^1, \mathbf{x}_k^2, ..., \mathbf{x}_k^N] \\
\mathbf{G}_k &\triangleq [\mathbf{g}_k^1, \mathbf{g}_k^2, ..., \mathbf{g}_k^N] \\
\mathbf{H}_k &\triangleq [\nabla f_1(\mathbf{x}_k^1), \nabla f_2(\mathbf{x}_k^2), ..., \nabla f_N(\mathbf{x}_k^N)]
\end{aligned}
\tag{16}
$$

We can observe that the above matrices are all with dimension $d \times N$ such that any matrix $\mathbf{A}$ satisfies $\|\mathbf{A}\|_{\mathfrak{F}}^2 = \sum_{i=1}^N \|\mathbf{a}_i\|^2$, where $\mathbf{a}_i$ is the $i$-th column of the matrix $\mathbf{A}$. Thus, we can obtain that:

$$
\|\mathbf{X}_k(\mathbf{I} - \mathbf{Q})\|_{\mathfrak{F}}^2 = \sum_{i=1}^N \|\mathbf{x}_k^i - \bar{\mathbf{x}}_k\|^2.
\tag{17}
$$

Define $\mathbf{Q} = \frac{1}{N}\mathbf{1}\mathbf{1}^\top$. For each doubly stochastic matrix $\mathbf{W}$, the following properties can be obtained

$$
\begin{aligned}
(i) \quad & \mathbf{Q}\mathbf{W} = \mathbf{W}\mathbf{Q}; \\
(ii) \quad & (\mathbf{I} - \mathbf{Q})\mathbf{W} = \mathbf{W}(\mathbf{I} - \mathbf{Q}); \\
(iii) \quad & \text{For any integer } k \geq 1, \|(\mathbf{I} - \mathbf{Q})\mathbf{W}\|_{\mathfrak{S}} \leq (\sqrt{\rho})^k, \text{ where } \|\cdot\|_{\mathfrak{S}} \text{ is the spectrum norm of a matrix.}
\end{aligned}
\tag{18}
$$

Let $\mathbf{A}_i, i \in \{1, 2, ..., N\}$ be $N$ arbitrary real square matrices. It follows that

$$
\|\sum_{i=1}^N \mathbf{A}_i\|_{\mathfrak{F}}^2 \leq \sum_{i=1}^N \sum_{j=1}^N \|\mathbf{A}_i\|_{\mathfrak{F}}\|\mathbf{A}_j\|_{\mathfrak{F}}.
\tag{19}
$$

We present the following auxiliary lemmas to prove the lemma. 2

**Lemma 3.** *Let all assumptions hold. Let $g^i$ be the unbiased estimate of $\nabla f_i(\mathbf{x}^i)$ at the point $\mathbf{x}^i$ such that $\mathbb{E}[\mathbf{g}^i] = \nabla f_i(\mathbf{x}^i)$, for all $i \in [N] := \{1, 2, ..., N\}$. Thus the following relationship holds*

$$
\mathbb{E}\left[\left\|\frac{1}{N}\sum_{i=1}^N \tilde{\mathbf{g}}^i\right\|^2\right] \leq \frac{10\sigma^2}{N} + 8\zeta^2 + 2\mathbb{E}\left[\left\|\frac{1}{N}\sum_{i=1}^N \nabla f_i(\mathbf{x}^i)\right\|^2\right]
\tag{20}
$$

*Proof for Lemma 3:*

$$
\begin{aligned}
\mathbb{E}\left[\left\|\frac{1}{N}\sum_{i=1}^N \tilde{\mathbf{g}}^i\right\|^2\right] &= \mathbb{E}\left[\left\|\frac{1}{N}\sum_{i=1}^N (\tilde{\mathbf{g}}^i - \mathbf{g}^i + \mathbf{g}^i)\right\|^2\right] \\
&= \mathbb{E}\left[\left\|\frac{1}{N}\sum_{i=1}^N (\tilde{\mathbf{g}}^i - \mathbf{g}^i) + \frac{1}{N}\sum_{i=1}^N \mathbf{g}^i\right\|^2\right] \\
&\overset{a}{\leq} 2\mathbb{E}\left[\left\|\frac{1}{N}\sum_{i=1}^N (\tilde{\mathbf{g}}^i - \mathbf{g}^i)\right\|^2\right] + 2\mathbb{E}\left[\left\|\frac{1}{N}\sum_{i=1}^N \mathbf{g}^i\right\|^2\right] \\
&\overset{b}{\leq} 8\left(\frac{\sigma^2}{N} + \zeta^2\right) + \frac{2\sigma^2}{N} + 2\mathbb{E}\left[\left\|\frac{1}{N}\sum_{i=1}^N \nabla f_i(\mathbf{x}^i)\right\|^2\right] \\
&= \frac{10\sigma^2}{N} + 8\zeta^2 + 2\mathbb{E}\left[\left\|\frac{1}{N}\sum_{i=1}^N \nabla f_i(\mathbf{x}^i)\right\|^2\right]
\end{aligned}
$$

(a) refers to the fact that the inequality $\|\mathbf{a} + \mathbf{b}\|^2 \leq 2\|\mathbf{a}\|^2 + 2\|\mathbf{b}\|^2$. (b) The first term uses Lemma 1 and the second term uses the conclusion of Lemma 1 in (Yu et al., 2019).

**Lemma 4.** *Let all assumptions hold. Let $g^i$ be the unbiased estimate of $\nabla f_i(\mathbf{x}^i)$ at the point $\mathbf{x}^i$ such that $\mathbb{E}[\mathbf{g}^i] = \nabla f_i(\mathbf{x}^i)$, for all $i \in [N] := \{1, 2, ..., N\}$. Thus the following relationship holds*

$$\mathbb{E}\left[\left\|\tilde{\mathbf{G}}_\tau - \mathbf{G}_\tau\right\|_{\mathfrak{F}}^2\right] \leq 4N(\sigma^2 + \zeta^2) \tag{21}$$

*Proof for Lemma 4 is similar to Lemma 1 and is as follows:*

$$\mathbb{E}\left[\left\|\tilde{\mathbf{G}}_\tau - \mathbf{G}_\tau\right\|_{\mathfrak{F}}^2\right] = \sum_{i=1}^{N} \mathbb{E}\left[\left\|\sum_{j \in \mathcal{N}_i} w_{ij}(\nabla F_j(x, d^j) - \nabla F_i(x, d^i))\right\|^2\right]$$

$$= \sum_{i=1}^{N} \mathbb{E}\left[\left\|\sum_{j \in \mathcal{N}_i} w_{ij}(\nabla F_j(x, d^j) - \nabla f_j(x)) - (\nabla F_i(x, d^i) - \nabla f_i(x))\right\|^2\right]$$

$$+ \sum_{i=1}^{N} \mathbb{E}\left[\left\|\sum_{j \in \mathcal{N}_i} w_{ij}(\nabla f_j(x) - \nabla f_i(x))\right\|^2\right]$$

$$\leq \sum_{i=1}^{N} \sum_{j \in \mathcal{N}_i} w_{ij}\mathbb{E}\left[\left\|\nabla F_j(x, d^j) - \nabla f_j(x) - (\nabla F_i(x, d^i) - \nabla f_i(x))\right\|^2\right]$$

$$+ \sum_{i=1}^{N} \sum_{j \in \mathcal{N}_i} w_{ij}\mathbb{E}\left[\left\|\nabla f_j(x) - \nabla \mathcal{F}(x) + \nabla \mathcal{F}(x) - \nabla f_i(x)\right\|^2\right]$$

$$\leq 4N(\sigma^2 + \zeta^2) \quad \text{(follows from assumption 2)}$$

The properties shown in Eq. 18 and 19 have been well established and in this context, we skip the proof here. We are now ready to prove Lemma 2.

*Proof for Lemma 2:* Since $\mathbf{X}_k = \mathbf{X}_{k-1}\mathbf{W} + \mathbf{V}_k$ we have:

$$\mathbf{X}_k(\mathbf{I} - \mathbf{Q}) = \mathbf{X}_{k-1}(\mathbf{I} - \mathbf{Q})\mathbf{W} + \mathbf{V}_k(\mathbf{I} - \mathbf{Q}) \tag{22}$$

Applying the above equation $k$ times we have:

$$\mathbf{X}_k(\mathbf{I} - \mathbf{Q}) = \mathbf{X}_0(\mathbf{I} - \mathbf{Q})\mathbf{W}^k + \sum_{\tau=1}^{k} \mathbf{V}_\tau(\mathbf{I} - \mathbf{Q})\mathbf{W}^{k-\tau} \stackrel{\mathbf{X}_0=0}{=} \sum_{\tau=1}^{k} \mathbf{V}_\tau(\mathbf{I} - \mathbf{Q})\mathbf{W}^{k-\tau} \tag{23}$$

As $\bar{\mathbf{V}}_k = \beta\bar{\mathbf{V}}_{k-1} - \eta\frac{1}{N}\sum_{i=1}^{N} \tilde{\mathbf{G}}_{k-1}^i \stackrel{\mathbf{V}_0=0}{=} -\eta\frac{1}{N}\sum_{i=1}^{N} \tilde{\mathbf{G}}_{k-1}^i$, we can get:

$$\mathbf{X}_k(\mathbf{I} - \mathbf{Q}) = -\eta \sum_{\tau=1}^{k} \sum_{l=0}^{\tau-1} \tilde{\mathbf{G}}_l \beta^{\tau-1-l}(\mathbf{I} - \mathbf{Q})\mathbf{W}^{k-\tau}$$

$$= -\eta \sum_{\tau=1}^{k} \sum_{l=0}^{\tau-1} \tilde{\mathbf{G}}_l \beta^{\tau-1-l}\mathbf{W}^{k-\tau}(\mathbf{I} - \mathbf{Q}) - \eta \sum_{n=1}^{k-1} \tilde{\mathbf{G}}_n [\sum_{l=n+1}^{k} \beta^{l-1-n}\mathbf{W}^{k-1-n}(\mathbf{I} - \mathbf{Q}) \tag{24}$$

$$= -\eta \sum_{\tau=0}^{k-1} \frac{1 - \beta^{k-\tau}}{1 - \beta} \tilde{\mathbf{G}}_\tau(\mathbf{I} - \mathbf{Q})\mathbf{W}^{k-1-\tau}.$$

Therefore, for $k \geq 1$, we have:

$$\mathbb{E}\left[\left\|\mathbf{X}_k(\mathbf{I} - \mathbf{Q})\right\|_{\mathfrak{F}}^2\right] = \eta^2 \mathbb{E}\left[\left\|\sum_{\tau=0}^{k-1} \frac{1 - \beta^{k-\tau}}{1 - \beta} \tilde{\mathbf{G}}_\tau (\mathbf{I} - \mathbf{Q}) \mathbf{W}^{k-1-\tau}\right\|_{\mathfrak{F}}^2\right]$$

$$\overset{a}{\leq} \underbrace{2\eta^2 \mathbb{E}\left[\left\|\sum_{\tau=0}^{k-1} \frac{1 - \beta^{k-\tau}}{1 - \beta} (\tilde{\mathbf{G}}_\tau - \mathbf{G}_\tau)(\mathbf{I} - \mathbf{Q}) \mathbf{W}^{k-1-\tau}\right\|_{\mathfrak{F}}^2\right]}_{I} +$$

$$\underbrace{2\eta^2 \mathbb{E}\left[\left\|\sum_{\tau=0}^{k-1} \frac{1 - \beta^{k-\tau}}{1 - \beta} \mathbf{G}_\tau (\mathbf{I} - \mathbf{Q}) \mathbf{W}^{k-1-\tau}\right\|_{\mathfrak{F}}^2\right]}_{II} \tag{25}$$

(a) follows from the inequality $\|\mathbf{A} + \mathbf{B}\|_{\mathfrak{F}}^2 \leq 2\|\mathbf{A}\|_{\mathfrak{F}}^2 + 2\|\mathbf{B}\|_{\mathfrak{F}}^2$.

We develop upper bounds of the term $\mathbf{I}$:

$$\mathbb{E}\left[\left\|\sum_{\tau=0}^{k-1} \frac{1 - \beta^{k-\tau}}{1 - \beta} (\tilde{\mathbf{G}}_\tau - \mathbf{G}_\tau)(\mathbf{I} - \mathbf{Q}) \mathbf{W}^{k-1-\tau}\right\|_{\mathfrak{F}}^2\right] \overset{a}{\leq} \sum_{\tau=0}^{k-1} \mathbb{E}\left[\left\|\frac{1 - \beta^{k-\tau}}{1 - \beta} (\tilde{\mathbf{G}}_\tau - \mathbf{G}_\tau)(\mathbf{I} - \mathbf{Q}) \mathbf{W}^{k-1-\tau}\right\|_{\mathfrak{F}}^2\right]$$

$$\overset{b}{\leq} \frac{1}{(1 - \beta)^2} \sum_{\tau=0}^{k-1} \rho^{k-1-\tau} \mathbb{E}\left[\left\|\tilde{\mathbf{G}}_\tau - \mathbf{G}_\tau\right\|_{\mathfrak{F}}^2\right]$$

$$\overset{c}{\leq} \frac{4}{(1 - \beta)^2} \sum_{\tau=0}^{k-1} \rho^{k-1-\tau} N(\sigma^2 + \zeta^2) \overset{d}{\leq} \frac{4N(\sigma^2 + \zeta^2)}{(1 - \beta)^2(1 - \rho)} \tag{26}$$

(a) follows from Jensen's inequality. (b) follows from the inequality $\left|\frac{1-\beta^{k-\tau}}{1-\beta}\right| \leq \frac{1}{1-\beta}$. (c) follows from Lemma 2 and Frobenius norm. (d) follows from Assumption 3.

We then proceed to find the upper bound for term $\mathbf{II}$.

$$\mathbb{E}\left[\left\|\sum_{\tau=0}^{k-1} \frac{1 - \beta^{k-\tau}}{1 - \beta} \mathbf{G}_\tau (\mathbf{I} - \mathbf{Q}) \mathbf{W}^{k-1-\tau}\right\|_{\mathfrak{F}}^2\right] \overset{a}{\leq} \sum_{\tau=0}^{k-1} \sum_{\tau'=0}^{k-1} \mathbb{E}\left[\left\|\frac{1 - \beta^{k-\tau}}{1 - \beta} \mathbf{G}_\tau (\mathbf{I} - \mathbf{Q}) \mathbf{W}^{k-1-\tau}\right\|_{\mathfrak{F}}\right.$$

$$\left.\left\|\frac{1 - \beta^{k-\tau}}{1 - \beta} \mathbf{G}_{\tau'} (\mathbf{I} - \mathbf{Q}) \mathbf{W}^{k-1-\tau'}\right\|_{\mathfrak{F}}\right]$$

$$\leq \frac{1}{(1 - \beta)^2} \sum_{\tau=0}^{k-1} \sum_{\tau'=0}^{k-1} \rho^{(k-1-\frac{\tau+\tau'}{2})} \mathbb{E}\left[\|\mathbf{G}_\tau\|_{\mathfrak{F}} \|\mathbf{G}_{\tau'}\|_{\mathfrak{F}}\right]$$

$$\overset{b}{\leq} \frac{1}{(1 - \beta)^2} \sum_{\tau=0}^{k-1} \sum_{\tau'=0}^{k-1} \rho^{(k-1-\frac{\tau+\tau'}{2})} \left(\frac{1}{2}\mathbb{E}[\|\mathbf{G}_\tau\|_{\mathfrak{F}}^2] + \frac{1}{2}\mathbb{E}[\|\mathbf{G}_{\tau'}\|_{\mathfrak{F}}^2]\right)$$

$$= \frac{1}{(1 - \beta)^2} \sum_{\tau=0}^{k-1} \sum_{\tau'=0}^{k-1} \rho^{(k-1-\frac{\tau+\tau'}{2})} \mathbb{E}[\|\mathbf{G}_\tau\|_{\mathfrak{F}}^2]$$

$$\overset{c}{\leq} \frac{1}{(1 - \beta)^2(1 - \sqrt{\rho})} \sum_{\tau=0}^{k-1} \rho^{(\frac{k-1-\tau}{2})} \mathbb{E}[\|\mathbf{G}_\tau\|_{\mathfrak{F}}^2] \tag{27}$$

(a) follows from Eq. 19. (b) follows from the inequality $xy \leq \frac{1}{2}(x^2 + y^2)$ for any two real numbers $x, y$. (c) is derived using $\sum_{\tau_1=0}^{k-1} \rho^{k-1-\frac{\tau_1+\tau}{2}} \leq \frac{\rho^{\frac{k-1-\tau}{2}}}{1-\sqrt{\rho}}$.

We then proceed with finding the bounds for $\mathbb{E}[\|\mathbf{G}_\tau\|_{\mathfrak{F}}^2]$:

$$
\begin{aligned}
\mathbb{E}[\|\mathbf{G}_\tau\|_{\mathfrak{F}}^2] &= \mathbb{E}[\|\mathbf{G}_\tau - \mathbf{H}_\tau + \mathbf{H}_\tau - \mathbf{H}_\tau \mathbf{Q} + \mathbf{H}_\tau \mathbf{Q}\|_{\mathfrak{F}}^2] \\
&\leq 3\mathbb{E}[\|\mathbf{G}_\tau - \mathbf{H}_\tau\|_{\mathfrak{F}}^2] + 3\mathbb{E}[\|\mathbf{H}_\tau(I - \mathbf{Q})\|_{\mathfrak{F}}^2] + 3\mathbb{E}[\|\mathbf{H}_\tau \mathbf{Q}\|_{\mathfrak{F}}^2] \\
&\overset{a}{\leq} 3N\sigma^2 + 3N\zeta^2 + 3\mathbb{E}[\|\frac{1}{N}\sum_{i=1}^N \nabla f_i(\mathbf{x}_\tau^i)\|^2]
\end{aligned}
\tag{28}
$$

(a) holds because $\mathbb{E}[\|\mathbf{H}_\tau \mathbf{Q}\|_{\mathfrak{F}}^2] \leq \mathbb{E}[\|\frac{1}{N}\sum_{i=1}^N \nabla f_i(\mathbf{x}_\tau^i)\|^2]$

Substituting (28) in (27):

$$
\begin{aligned}
\mathbb{E}\left[\left\|\sum_{\tau=0}^{k-1} \frac{1-\beta^{k-\tau}}{1-\beta}\mathbf{G}_\tau(\mathbf{I} - \mathbf{Q})\mathbf{W}^{k-1-\tau}\right\|_{\mathfrak{F}}^2\right] &\leq \frac{1}{(1-\beta)^2(1-\sqrt{\rho})}\sum_{\tau=0}^{k-1}\rho^{(\frac{k-1-\tau}{2})}\left[3N\sigma^2 + 3N\zeta^2 + 3N\mathbb{E}[\|\frac{1}{N}\sum_{i=1}^N \nabla f_i(\mathbf{x}_\tau^i)\|^2]\right] \\
&\leq \frac{3N(\sigma^2+\zeta^2)}{(1-\beta)^2(1-\sqrt{\rho})^2} + \frac{3N}{(1-\beta)^2(1-\sqrt{\rho})}\sum_{\tau=0}^{k-1}\rho^{(\frac{k-1-\tau}{2})}\mathbb{E}[\|\frac{1}{N}\sum_{i=1}^N \nabla f_i(\mathbf{x}_\tau^i)\|^2]
\end{aligned}
\tag{29}
$$

substituting (29) and (26) into the main inequality (25):

$$
\begin{aligned}
\mathbb{E}\left[\left\|\mathbf{X}_k(\mathbf{I}-\mathbf{Q})\right\|_{\mathfrak{F}}^2\right] &\leq \frac{8\eta^2 N(\sigma^2+\zeta^2)}{(1-\beta)^2(1-\rho)} + \frac{2\eta^2}{(1-\beta)^2(1-\sqrt{\rho})}\left(\frac{3N(\sigma^2)}{1-\sqrt{\rho}} + \frac{3N(\zeta^2)}{1-\sqrt{\rho}} + \right. \\
&\qquad\qquad\qquad\qquad\qquad\qquad\qquad \left. 3N\sum_{\tau=0}^{k-1}\rho^{(\frac{k-1-\tau}{2})}\mathbb{E}[\|\frac{1}{N}\sum_{i=1}^N \nabla f_i(\mathbf{x}_\tau^i)\|^2]\right) \\
&= \frac{2\eta^2}{(1-\beta)^2}\left(\frac{4N(\sigma^2+\zeta^2)}{1-\rho} + \frac{3N\sigma^2}{(1-\sqrt{\rho})^2} + \frac{3N\zeta^2}{(1-\sqrt{\rho})^2}\right) + \\
&\qquad\qquad \frac{6N\eta^2}{(1-\beta)^2(1-\sqrt{\rho})}\sum_{\tau=0}^{k-1}\rho^{(\frac{k-1-\tau}{2})}\mathbb{E}[\|\frac{1}{N}\sum_{i=1}^N \nabla f_i(\mathbf{x}_\tau^i)\|^2] \\
&\leq \frac{14N\eta^2}{(1-\beta)^2}\left(\frac{\sigma^2}{(1-\sqrt{\rho})^2} + \frac{\zeta^2}{(1-\sqrt{\rho})^2}\right) + \frac{6N\eta^2}{(1-\beta)^2(1-\sqrt{\rho})}\sum_{\tau=0}^{k-1}\rho^{(\frac{k-1-\tau}{2})}\mathbb{E}[\|\frac{1}{N}\sum_{i=1}^N \nabla f_i(\mathbf{x}_\tau^i)\|^2]
\end{aligned}
\tag{30}
$$

Summing over $k \in \{1, \ldots, K-1\}$ and noting that $\mathbb{E}\left[\left\|\mathbf{X}_0(\mathbf{I}-\mathbf{Q})\right\|_{\mathfrak{F}}^2\right] = 0$:

$$
\begin{aligned}
\sum_{k=1}^{K-1}\mathbb{E}\left[\left\|\mathbf{X}_k(\mathbf{I}-\mathbf{Q})\right\|_{\mathfrak{F}}^2\right] &\leq CK + \frac{6N\eta^2}{(1-\beta)^2(1-\sqrt{\rho})}\sum_{k=1}^{K-1}\sum_{\tau=0}^{k-1}\rho^{(\frac{k-1-\tau}{2})}\mathbb{E}[\|\frac{1}{N}\sum_{i=1}^N \nabla f_i(\mathbf{x}_\tau^i)\|^2] \\
&\leq CK + \frac{6N\eta^2}{(1-\beta)^2(1-\sqrt{\rho})}\sum_{k=0}^{K-1}\frac{1-\rho^{(\frac{K-1-k}{2})}}{1-\sqrt{\rho}}\mathbb{E}[\|\frac{1}{N}\sum_{i=1}^N \nabla f_i(\mathbf{x}_k^i)\|^2] \\
&\leq CK + \frac{6N\eta^2}{(1-\beta)^2(1-\sqrt{\rho})}\sum_{k=0}^{K-1}\mathbb{E}[\|\frac{1}{N}\sum_{i=1}^N \nabla f_i(\mathbf{x}_k^i)\|^2]
\end{aligned}
\tag{31}
$$

Where $C = \frac{14N\eta^2}{(1-\beta)^2}\left(\frac{\sigma^2}{(1-\sqrt{\rho})^2} + \frac{\zeta^2}{(1-\sqrt{\rho})^2}\right)$.

Dividing both sides by $N$:

$$\sum_{k=1}^{K-1}\frac{1}{N}\mathbb{E}\left[\left\|\mathbf{X}_k(\mathbf{I}-\mathbf{Q})\right\|_{\mathfrak{F}}^2\right] \leq \frac{14\eta^2}{(1-\beta)^2}\left(\frac{\sigma^2}{(1-\sqrt{\rho})^2} + \frac{\zeta^2}{(1-\sqrt{\rho})^2}\right)K +$$
$$\frac{6\eta^2}{(1-\beta)^2(1-\sqrt{\rho})}\sum_{k=0}^{K-1}\mathbb{E}[\|\frac{1}{N}\sum_{i=1}^N \nabla f_i(\mathbf{x}_k^i)\|^2] \tag{32}$$

Hence, we obtain the following as the lemma. 2:

$$\sum_{k=0}^{K-1}\frac{1}{N}\sum_{i=1}^N \mathbb{E}\left[\left\|\bar{\mathbf{x}}_k - \mathbf{x}_k^i\right\|^2\right] \leq \frac{14\eta^2(\sigma^2+\zeta^2)}{(1-\beta)^2(1-\sqrt{\rho})^2}K + \frac{6\eta^2}{(1-\beta)^2(1-\sqrt{\rho})}\sum_{k=0}^{K-1}\mathbb{E}[\|\frac{1}{N}\sum_{i=1}^N \nabla f_i(\mathbf{x}_k^i)\|^2] \tag{33}$$

### A.2.2 Proof for Theorem. 1

*Proof:* Using the L-smoothness properties for $\mathcal{F}$ we have:

$$\mathbb{E}[\mathcal{F}(\bar{\mathbf{z}}_{k+1})] \leq \mathbb{E}[\mathcal{F}(\bar{\mathbf{z}}_k)] + \mathbb{E}[\langle\nabla\mathcal{F}(\bar{\mathbf{z}}_k),\bar{\mathbf{z}}_{k+1}-\bar{\mathbf{z}}_k\rangle] + \frac{L}{2}\mathbb{E}[\|\bar{\mathbf{z}}_{k+1}-\bar{\mathbf{z}}_k\|^2] \tag{34}$$

Using Eq. 14 we have:

$$\mathbb{E}[\langle\nabla\mathcal{F}(\bar{\mathbf{z}}_k),\bar{\mathbf{z}}_{k+1}-\bar{\mathbf{z}}_k\rangle] = \frac{-\eta}{1-\beta}\mathbb{E}[\langle\nabla\mathcal{F}(\bar{\mathbf{z}}_k),\frac{1}{N}\sum_{i=1}^N \tilde{\mathbf{g}}_k^i\rangle] =$$
$$\underbrace{\frac{-\eta}{1-\beta}\mathbb{E}[\langle\nabla\mathcal{F}(\bar{\mathbf{z}}_k)-\nabla\mathcal{F}(\bar{\mathbf{x}}_k),\frac{1}{N}\sum_{i=1}^N \tilde{\mathbf{g}}_k^i\rangle]}_{T_1} \underbrace{-\frac{\eta}{1-\beta}\mathbb{E}[\langle\nabla\mathcal{F}(\bar{\mathbf{x}}_k),\frac{1}{N}\sum_{i=1}^N \tilde{\mathbf{g}}_k^i\rangle]}_{T_2} \tag{35}$$

We proceed by bounding $T_1$:

$$\frac{-\eta}{1-\beta}\mathbb{E}[\langle\nabla\mathcal{F}(\bar{\mathbf{z}}_k)-\nabla\mathcal{F}(\bar{\mathbf{x}}_k),\frac{1}{N}\sum_{i=1}^N \tilde{\mathbf{g}}_k^i\rangle]$$
$$\overset{(i)}{\leq} \frac{(1-\beta)}{2\beta L}\mathbb{E}[\|\nabla\mathcal{F}(\bar{\mathbf{z}}_k)-\nabla\mathcal{F}(\bar{\mathbf{x}}_k)\|^2] + \frac{\beta L\eta^2}{2(1-\beta)^3}\mathbb{E}[\|\frac{1}{N}\sum_{i=1}^N \tilde{\mathbf{g}}_k^i\|^2] \tag{36}$$
$$\overset{(ii)}{\leq} \frac{(1-\beta)L}{2\beta}\mathbb{E}[\|\bar{\mathbf{z}}_k-\bar{\mathbf{x}}_k\|^2] + \frac{\beta L\eta^2}{2(1-\beta)^3}\mathbb{E}[\|\frac{1}{N}\sum_{i=1}^N \tilde{\mathbf{g}}_k^i\|^2]$$

(i) holds as $\langle\mathbf{a},\mathbf{b}\rangle \leq \frac{1}{2}\|\mathbf{a}\|^2 + \frac{1}{2}\|\mathbf{b}\|^2$ where $\mathbf{a} = \frac{\sqrt{1-\beta}}{\sqrt{\beta L}}(\nabla\mathcal{F}(\bar{\mathbf{z}}_k)-\nabla\mathcal{F}(\bar{\mathbf{x}}_k))$ and $\mathbf{b} = -\frac{\eta\sqrt{\beta L}}{(1-\beta)^{\frac{3}{2}}}\frac{1}{N}\sum_{i=1}^N \tilde{\mathbf{g}}_k^i$. and (ii) uses the fact that $\mathcal{F}$ is L-smooth.

We split the term $T_2$ as follows:

$$\langle \nabla \mathcal{F}\left(\bar{\mathbf{x}}_k\right), \frac{1}{N}\sum_{i=1}^{N}\tilde{\mathbf{g}}_k^i\rangle = \langle \nabla \mathcal{F}\left(\bar{\mathbf{x}}_k\right), \frac{1}{N}\sum_{i=1}^{N}\left(\tilde{\mathbf{g}}_k^i - \mathbf{g}_k^i + \mathbf{g}_k^i\right)\rangle =$$

$$\underbrace{\langle \nabla \mathcal{F}\left(\bar{\mathbf{x}}_k\right), \frac{1}{N}\sum_{i=1}^{N}\left(\tilde{\mathbf{g}}_k^i - \mathbf{g}_k^i\right)\rangle}_{T_3} + \underbrace{\langle \nabla \mathcal{F}\left(\bar{\mathbf{x}}_k\right), \frac{1}{N}\sum_{i=1}^{N}\mathbf{g}_k^i\rangle}_{T_4} \tag{37}$$

Now, We first analyze the term $T_3$:

$$\frac{-\eta}{(1-\beta)}\mathbb{E}[\langle \nabla \mathcal{F}\left(\bar{\mathbf{x}}_k\right), \frac{1}{N}\sum_{i=1}^{N}\left(\tilde{\mathbf{g}}_k^i - \mathbf{g}_k^i\right)\rangle]$$

$$\leq \frac{(1-\beta)}{2\beta L}\mathbb{E}[\|\nabla \mathcal{F}(\bar{\mathbf{x}}_k)\|^2] + \frac{\eta^2 \beta L}{2(1-\beta)^3}\mathbb{E}[\|\frac{1}{N}\sum_{i=1}^{N}(\tilde{\mathbf{g}}_k^i - \mathbf{g}_k^i)\|^2] \tag{38}$$

This holds as $\langle \mathbf{a}, \mathbf{b}\rangle \leq \frac{1}{2}\|\mathbf{a}\|^2 + \frac{1}{2}\|\mathbf{b}\|^2$ where $\mathbf{a} = \frac{-\sqrt{1-\beta}}{\sqrt{\beta L}}\nabla \mathcal{F}(\bar{\mathbf{x}}_k)$ and $\mathbf{b} = \frac{\eta\sqrt{\beta L}}{(1-\beta)^{\frac{3}{2}}}\frac{1}{N}\sum_{i=1}^{N}(\tilde{\mathbf{g}}_k^i - \mathbf{g}_k^i)$.

Analyzing the term $T_4$:

$$\mathbb{E}\left[\langle \nabla \mathcal{F}\left(\bar{\mathbf{x}}_k\right), \frac{1}{N}\sum_{i=1}^{N}\mathbf{g}_k^i\rangle\right] = \mathbb{E}\left[\langle \nabla \mathcal{F}(\bar{\mathbf{x}}_k), \frac{1}{N}\sum_{i=1}^{N}\nabla f_i(\mathbf{x}_k^i)\rangle\right] \tag{39}$$

Using the equity $\langle \mathbf{a}, \mathbf{b}\rangle = \frac{1}{2}[\|\mathbf{a}\|^2 + \|\mathbf{b}\|^2 - \|\mathbf{a} - \mathbf{b}\|^2]$, we have :

$$\langle \nabla \mathcal{F}\left(\bar{\mathbf{x}}_k\right), \frac{1}{N}\sum_{i=1}^{N}\nabla f_i\left(\mathbf{x}_k^i\right)\rangle = \frac{1}{2}\left(\|\nabla \mathcal{F}\left(\bar{\mathbf{x}}_k\right)\|^2 + \|\frac{1}{N}\sum_{i=1}^{N}\nabla f_i(\mathbf{x}_k^i)\|^2 - \|\nabla \mathcal{F}(\bar{\mathbf{x}}_k) - \frac{1}{N}\sum_{i=1}^{N}\nabla f_i(\mathbf{x}_k^i)\|^2\right)$$

$$\overset{a}{\geq} \frac{1}{2}\left(\|\nabla \mathcal{F}(\bar{\mathbf{x}}_k)\|^2 + \|\frac{1}{N}\sum_{i=1}^{N}\nabla f_i(\mathbf{x}_k^i)\|^2 - L^2 \frac{1}{N}\sum_{i=1}^{N}\|\bar{\mathbf{x}}_k - \mathbf{x}_k^i\|^2\right) \tag{40}$$

(a) holds because $\|\nabla \mathcal{F}(\bar{\mathbf{x}}_k) - \frac{1}{N}\sum_{i=1}^{N}\nabla f_i(\mathbf{x}_k^i)\|^2 = \|\frac{1}{N}\sum_{i=1}^{N}\nabla f_i(\bar{\mathbf{x}}_k) - \frac{1}{N}\sum_{i=1}^{N}\nabla f_i(\mathbf{x}_k^i)\|^2 \leq \frac{1}{N}\sum_{i=1}^{N}\|\nabla f_i(\bar{\mathbf{x}}_k) - \nabla f_i(\mathbf{x}_k^i)\|^2 \leq \frac{1}{N}\sum_{i=1}^{N}L^2\|\bar{\mathbf{x}}_k - \mathbf{x}_k^i\|^2$.

Substituting (40) into (39), we have

$$\mathbb{E}\left[\langle \nabla \mathcal{F}\left(\bar{\mathbf{x}}_k\right), \frac{1}{N}\sum_{i=1}^{N}\mathbf{g}_k^i\rangle\right] \geq \frac{1}{2}\left(\mathbb{E}[\|\nabla \mathcal{F}(\bar{\mathbf{x}}_k)\|^2] + \mathbb{E}[\|\frac{1}{N}\sum_{i=1}^{N}\nabla f_i(\mathbf{x}_k^i)\|^2] - L^2 \frac{1}{N}\sum_{i=1}^{N}\mathbb{E}[\|\bar{\mathbf{x}}_k - \mathbf{x}_k^i\|^2]\right) \tag{41}$$

Substituting (38), (41) into (37), we have

$$-\frac{\eta}{1-\beta}\mathbb{E}[\langle \nabla \mathcal{F}(\bar{\mathbf{x}}_k), \frac{1}{N}\sum_{i=1}^{N}\tilde{\mathbf{g}}_k^i\rangle] \leq \left(\frac{(1-\beta)}{2\beta L} - \frac{\eta}{2(1-\beta)}\right)\mathbb{E}[\|\nabla \mathcal{F}(\bar{\mathbf{x}}_k)\|^2] + \frac{\eta^2 \beta L}{2(1-\beta)^3}\mathbb{E}[\|\frac{1}{N}\sum_{i=1}^{N}(\tilde{\mathbf{g}}_k^i - \mathbf{g}_k^i)\|^2]$$

$$- \frac{\eta}{2(1-\beta)}\mathbb{E}[\|\frac{1}{N}\sum_{i=1}^{N}\nabla f_i(\mathbf{x}_k^i)\|^2] + \frac{\eta L^2}{2(1-\beta)}\frac{1}{N}\sum_{i=1}^{N}\mathbb{E}[\|\bar{\mathbf{x}}_k - \mathbf{x}_k^i\|^2] \tag{42}$$

Now, finally substituting (42), (36) into (35), we have

$$
\mathbb{E}[\langle \nabla \mathcal{F}(\bar{\mathbf{z}}_k), \bar{\mathbf{z}}_{k+1} - \bar{\mathbf{z}}_k \rangle] \leq \frac{(1-\beta)L}{2\beta} \mathbb{E}[\|\bar{\mathbf{z}}_k - \bar{\mathbf{x}}_k\|^2] + \frac{\beta L \eta^2}{2(1-\beta)^3} \mathbb{E}[\|\frac{1}{N}\sum_{i=1}^{N}(\tilde{\mathbf{g}}_k^i)\|^2] + \left( \frac{(1-\beta)}{2\beta L} - \frac{\eta}{2(1-\beta)} \right)
$$

$$
\mathbb{E}[\|\nabla \mathcal{F}(\bar{\mathbf{x}}_k)\|^2] - \frac{\eta}{2(1-\beta)} \mathbb{E}[\|\frac{1}{N}\sum_{i=1}^{N}\nabla f_i(\mathbf{x}_k^i)\|^2] + \frac{\eta^2 \beta L}{2(1-\beta)^3} \mathbb{E}[\|\frac{1}{N}\sum_{i=1}^{N}(\tilde{\mathbf{g}}_k^i - \mathbf{g}_k^i)\|^2] + \frac{\eta L^2}{2(1-\beta)} \frac{1}{N}\sum_{i=1}^{N}\mathbb{E}[\|\bar{\mathbf{x}}_k - \mathbf{x}_k^i\|^2]
$$

(43)

Equation. 14 states that:

$$
\mathbb{E}[\|\bar{\mathbf{z}}_{k+1} - \bar{\mathbf{z}}_k\|^2] = \frac{\eta^2}{(1-\beta)^2} \mathbb{E}[\|\frac{1}{N}\sum_{i=1}^{N}\tilde{\mathbf{g}}_k^i\|^2].
$$

(44)

Substituting (43), (44) in (34):

$$
\mathbb{E}[\mathcal{F}(\bar{\mathbf{z}}_{k+1})] \leq \mathbb{E}[\mathcal{F}(\bar{\mathbf{z}}_k)] + \frac{(1-\beta)L}{2\beta} \mathbb{E}[\|\bar{\mathbf{z}}_k - \bar{\mathbf{x}}_k\|^2] + \frac{\beta L \eta^2}{2(1-\beta)^3} \mathbb{E}[\|\frac{1}{N}\sum_{i=1}^{N}(\tilde{\mathbf{g}}_k^i)\|^2] + \left( \frac{(1-\beta)}{2\beta L} - \frac{\eta}{2(1-\beta)} \right)
$$

$$
\mathbb{E}[\|\nabla \mathcal{F}(\bar{\mathbf{x}}_k)\|^2] - \frac{\eta}{2(1-\beta)} \mathbb{E}[\|\frac{1}{N}\sum_{i=1}^{N}\nabla f_i(\mathbf{x}_k^i)\|^2] + \frac{\eta^2 \beta L}{2(1-\beta)^3} \mathbb{E}[\|\frac{1}{N}\sum_{i=1}^{N}(\tilde{\mathbf{g}}_k^i - \mathbf{g}_k^i)\|^2]+
$$

$$
\frac{\eta L^2}{2(1-\beta)} \frac{1}{N}\sum_{i=1}^{N}\mathbb{E}[\|\bar{\mathbf{x}}_k - \mathbf{x}_k^i\|^2] + \frac{\eta^2}{(1-\beta)^2} \mathbb{E}[\|\frac{1}{N}\sum_{i=1}^{N}\tilde{\mathbf{g}}_k^i\|^2].
$$

We find the bound for $\mathbb{E}[\|\nabla \mathcal{F}(\bar{\mathbf{x}}_k)\|^2]$ by rearranging the terms and dividing by $C_1 = \frac{\eta}{2(1-\beta)} - \frac{(1-\beta)}{2\beta L}$:

$$
\mathbb{E}[\|\nabla \mathcal{F}(\bar{\mathbf{x}}_k)\|^2] \leq \frac{1}{C_1} \left( \mathbb{E}[\mathcal{F}(\bar{\mathbf{z}}_k)] - \mathbb{E}[\mathcal{F}(\bar{\mathbf{z}}_{k+1})] \right) + \widetilde{C}_2 \, \mathbb{E}[\|\frac{1}{N}\sum_{i=1}^{N}(\tilde{\mathbf{g}}_k^i)\|^2] + \widetilde{C}_3 \, \mathbb{E}[\|\bar{\mathbf{z}}_k - \bar{\mathbf{x}}_k\|^2]
$$

$$
- \widetilde{C}_6 \, \mathbb{E}[\|\frac{1}{N}\sum_{i=1}^{N}\nabla f_i(\mathbf{x}_k^i)\|^2] + \widetilde{C}_4 \, \mathbb{E}[\|\frac{1}{N}\sum_{i=1}^{N}(\tilde{\mathbf{g}}_k^i - \mathbf{g}_k^i)\|^2] + \frac{\widetilde{C}_5}{N} \sum_{i=1}^{N}\mathbb{E}[\|\bar{\mathbf{x}}_k - \mathbf{x}_k^i\|^2]
$$

Where $\widetilde{C}_2 = \left( \frac{\beta L \eta^2}{2(1-\beta)^3} + \frac{\eta^2 L}{(1-\beta)^2} \right)/C_1$, $\widetilde{C}_3 = \frac{(1-\beta)L}{2\beta}/C_1$, $\widetilde{C}_4 = \frac{\eta^2 \beta L}{2(1-\beta)^3}/C_1$, $\widetilde{C}_5 = \frac{\eta L^2}{2(1-\beta)}/C_1$, $\widetilde{C}_6 = \frac{\eta}{2(1-\beta)}/C_1$.

Summing over $k \in \{0, 1, \ldots, K-1\}$:

$$
\sum_{k=0}^{K-1}\mathbb{E}\left[\|\nabla \mathcal{F}(\bar{\mathbf{x}}_k)\|^2\right] \leq \frac{1}{C_1}\left( \mathbb{E}\left[\mathcal{F}(\bar{\mathbf{z}}_0)\right] - \mathbb{E}\left[\mathcal{F}(\bar{\mathbf{z}}_k)\right] \right) - \widetilde{C}_6 \sum_{k=0}^{K-1}\mathbb{E}\left[\left\|\frac{1}{N}\sum_{i=1}^{N}\nabla f_i\left(\mathbf{x}_k^i\right)\right\|^2\right] + \widetilde{C}_2 \sum_{k=0}^{K-1}\mathbb{E}\left[\left\|\frac{1}{N}\sum_{i=1}^{N}\tilde{\mathbf{g}}_k^i\right\|^2\right]
$$

$$
+ \widetilde{C}_3 \sum_{k=0}^{K-1}\mathbb{E}\left[\|\bar{\mathbf{z}}_k - \bar{\mathbf{x}}_k\|^2\right] + \widetilde{C}_4 \sum_{k=0}^{K-1}\mathbb{E}\left[\left\|\frac{1}{N}\sum_{i=1}^{N}\left(\tilde{\mathbf{g}}_k^i - \mathbf{g}_k^i\right)\right\|^2\right] + \widetilde{C}_5 \sum_{k=0}^{K-1}\frac{1}{N}\sum_{l=1}^{N}\mathbb{E}\left[\|\bar{\mathbf{x}}_k - \mathbf{x}_k^i\|^2\right]
$$

Substituting Lemma 1, Lemma 2, Lemma 3 and Equation. 15 into the above equation, we have:

$$\sum_{k=0}^{K-1} \mathbb{E}\left[\|\nabla \mathcal{F}(\bar{\mathbf{x}}_k)\|^2\right] \le \frac{1}{C_1}\left(\mathbb{E}\left[\mathcal{F}(\bar{\mathbf{z}}_0)\right] - \mathbb{E}\left[\mathcal{F}(\bar{\mathbf{z}}_k)\right]\right) - \left(\widetilde{C}_6 - 2\widetilde{C}_2 - 2\widetilde{C}_3\frac{\eta^2\beta^2}{(1-\beta)^4} - \widetilde{C}_5\frac{6\eta^2}{(1-\beta)^2(1-\sqrt{\rho})}\right)$$

$$\sum_{k=0}^{K-1}\mathbb{E}\left[\left\|\frac{1}{N}\sum_{i=1}^{N}\nabla f_i(\mathbf{x}_k^i)\right\|^2\right] + \left(\widetilde{C}_2 + \widetilde{C}_3\frac{\eta^2\beta^2}{(1-\beta)^4}\right)\left(\frac{10\sigma^2}{N} + 8\zeta^2\right)K + 4\widetilde{C}_4\left(\frac{\sigma^2}{N} + \zeta^2\right)K$$

$$+ \widetilde{C}_5\frac{14\eta^2(\sigma^2+\zeta^2)}{(1-\beta)^2(1-\sqrt{\rho})^2}K$$

Dividing both sides by $K$ and considering the fact that $\bar{\mathbf{z}}_0 = \bar{\mathbf{x}}_0$ and $\left(\widetilde{C}_6 - 2\widetilde{C}_2 - 2\widetilde{C}_3\frac{\eta^2\beta^2}{(1-\beta)^4} - \widetilde{C}_5\frac{6\eta^2}{(1-\beta)^2(1-\sqrt{\rho})}\right) \ge 0$:

$$\frac{1}{K}\sum_{k=0}^{K-1}\mathbb{E}\left[\|\nabla\mathcal{F}(\bar{\mathbf{x}}_k)\|^2\right] \le \frac{1}{C_1 K}\left(\mathcal{F}(\bar{\mathbf{x}}_0) - \mathcal{F}^\star\right) + \left(\widetilde{C}_2 + \widetilde{C}_3\frac{\eta^2\beta^2}{(1-\beta)^4}\right)\left(\frac{10\sigma^2}{N} + 8\zeta^2\right)$$

$$+ 4\widetilde{C}_4\left(\frac{\sigma^2}{N} + \zeta^2\right) + \widetilde{C}_5\frac{14\eta^2(\sigma^2+\zeta^2)}{(1-\beta)^2(1-\sqrt{\rho})^2}$$

$$\le \frac{1}{C_1 K}(\mathcal{F}(\bar{\mathbf{x}}_0) - \mathcal{F}^\star) + \left(10\widetilde{C}_2 + 10\widetilde{C}_3\frac{\eta^2\beta^2}{(1-\beta)^4} + 4\widetilde{C}_4\right)\left(\frac{\sigma^2}{N} + \zeta^2\right) + \widetilde{C}_5\frac{14\eta^2(\sigma^2+\zeta^2)}{(1-\beta)^2(1-\sqrt{\rho})^2}$$

$$\le \frac{1}{C_1 K}(\mathcal{F}(\bar{x}^0) - \mathcal{F}^*) + \left(10\widetilde{C}_2 + 10\widetilde{C}_3\frac{\eta^2\beta^2}{(1-\beta)^4} + 4\widetilde{C}_4\right)\left(\frac{\sigma^2}{N} + \zeta^2\right) + \widetilde{C}_5\frac{14\eta^2(\sigma^2+\zeta^2)}{(1-\beta)^2(1-\sqrt{\rho})^2}$$

Now, we simplify the coefficients:

$$10\widetilde{C}_2 + 10\widetilde{C}_3\frac{\eta^2\beta^2}{(1-\beta)^4} + 4\widetilde{C}_4 \le \left(\frac{10\eta^2 L(\eta^2+\beta)}{(1-\beta)^3}\right)/C_1 = \frac{20\eta^2 L^2\beta(\eta^2+\beta)}{(1-\beta)^2(\eta\beta L - (1-\beta)^2)}$$

$$\widetilde{C}_5\frac{14\eta^2}{(1-\beta)^2} = \frac{14\eta^3 L^3\beta}{(1-\beta)^2(\eta\beta L - (1-\beta)^2)}$$

Now we redefine the coefficients:

$$C_2 = \frac{20\eta^2 L^2\beta(\eta^2+\beta)}{(1-\beta)^2(\eta\beta L - (1-\beta)^2)}$$

$$C_3 = \frac{14\eta^3 L^3\beta}{(1-\beta)^2(\eta\beta L - (1-\beta)^2)}$$

Therefore we arrive at the bound given by the theorem. 1:

$$\frac{1}{K}\sum_{k=0}^{K-1}\mathbb{E}[\|\nabla\mathcal{F}(\bar{x}_k)\|^2] \le \frac{1}{C_1 K}(\mathcal{F}(\bar{x}^0) - \mathcal{F}^*) + C_2(\frac{\sigma^2}{N} + \zeta^2) + C_3\frac{(\sigma^2+\zeta^2)}{(1-\sqrt{\rho})^2} \tag{45}$$

### A.2.3 Discussion on the Step Size

In the proof of Theorem 1, we assumed the following $\widetilde{C}_6 - 2\widetilde{C}_2 - 2\widetilde{C}_3\frac{\eta^2\beta^2}{(1-\beta)^4} - \widetilde{C}_5\frac{6\eta^2}{(1-\beta)^2(1-\sqrt{\rho})} \ge 0$. The above equation is true under the following conditions:

$$(i)\ \ C_1 > 0$$

$$(ii)\ \ 1 - \frac{4L}{(1-\beta)^2}\eta - \frac{6L^2}{(1-\beta)^2(1-\sqrt{\rho})}\eta^2 \ge 0$$

Solving the first inequality gives us $\beta > 1 - \frac{L\eta}{2}(\sqrt{1 + \frac{4}{L\eta}} - 1)$.

We can simply this further by using the Taylor series approximation i.e., $\frac{1}{L\eta} < \beta < 1$ if $\eta > \frac{4}{L}$

Now, solving the second inequality, combining the fact that $\eta > 0$, we have then the specific form of $\eta^*$

$$\eta^* = \frac{\sqrt{4(1 - \sqrt{\rho})^2 + 6(1 - \sqrt{\rho})(1 - \beta)^2} - 2(1 - \sqrt{\rho})}{6L}.$$

Therefore, the step size $\eta$ is defined as

$$\eta \leq \frac{\sqrt{4(1 - \sqrt{\rho})^2 + 6(1 - \sqrt{\rho})(1 - \beta)^2} - 2(1 - \sqrt{\rho})}{6L}$$

### A.2.4 Proof for Corollary 1

We assume that the step size $\eta$ is $\mathcal{O}(\frac{\sqrt{N}}{\sqrt{K}})$ and $\zeta^2$ is $\mathcal{O}(\frac{1}{\sqrt{K}})$. Given this assumption, we have the following

$$C_1 = \mathcal{O}(\frac{\sqrt{N}}{\sqrt{K}}), \widetilde{C}_2 = \mathcal{O}(\frac{\sqrt{N}}{\sqrt{K}}), \widetilde{C}_3 = \mathcal{O}(\frac{\sqrt{K}}{\sqrt{N}}), \widetilde{C}_4 = \mathcal{O}(\frac{\sqrt{N}}{\sqrt{K}}), \widetilde{C}_5 = \mathcal{O}(1),$$

This implies that

$$C_2 = \mathcal{O}(\frac{\sqrt{N}}{\sqrt{K}}), C_3 = \mathcal{O}(\frac{N}{K})$$

Now we proceed to find the order of each term in Equation. 45. To do that we first point out that

$$\frac{\mathcal{F}(\bar{\mathbf{x}}_0) - \mathcal{F}^*}{C_1 K} = \mathcal{O}\left(\frac{1}{\sqrt{NK}}\right).$$

For the remaining terms we have,

$$C_2 \frac{\sigma^2}{N} = \mathcal{O}\left(\frac{1}{\sqrt{NK}}\right), \quad C_2 \zeta^2 = \mathcal{O}\left(\frac{\sqrt{N}}{K}\right)$$

$$C_3 \frac{\sigma^2}{(1 - \sqrt{\rho})^2} = \mathcal{O}\left(\frac{N}{K}\right), \quad C_3 \frac{\zeta^2}{(1 - \sqrt{\rho})^2} = \mathcal{O}\left(\frac{N}{K^{1.5}}\right)$$

Therefore, by omitting the constant $N$ in this context, there exists a constant $C > 0$ such that the overall convergence rate is as follows:

$$\frac{1}{K} \sum_{k=0}^{K-1} \mathbb{E}\left[\|\nabla \mathcal{F}(\bar{\mathbf{x}}_k)\|^2\right] \leq C\left(\frac{1}{\sqrt{NK}} + \frac{1}{K} + \frac{1}{K^{1.5}}\right), \tag{46}$$

which suggests when $K$ is sufficiently large, $NGM$ enables the convergence rate of $\mathcal{O}(\frac{1}{\sqrt{NK}})$.

### A.3 Decentralized Learning Setup

The traditional decentralized learning algorithm (d-psgd) is described as Algorithm. 2. For the decentralized setup, we use an undirected ring and undirected torus graph topologies with a uniform mixing matrix. The undirected ring topology for any graph size has 3 peers per agent including itself and each edge has a weight of $\frac{1}{3}$. The undirected torus topology with 10 agents has 4 peers per agent including itself and each edge has a weight of $\frac{1}{4}$. The undirected torus topology 20 agents have 5 peers per agent including itself and each edge has a weight of $\frac{1}{5}$. Finally, Figure 5 depicts the overview of the proposed $NGM$ algorithm. Note that we do not have step 4 of the Figure 5 in $NGM_{mv}$ algorithm.

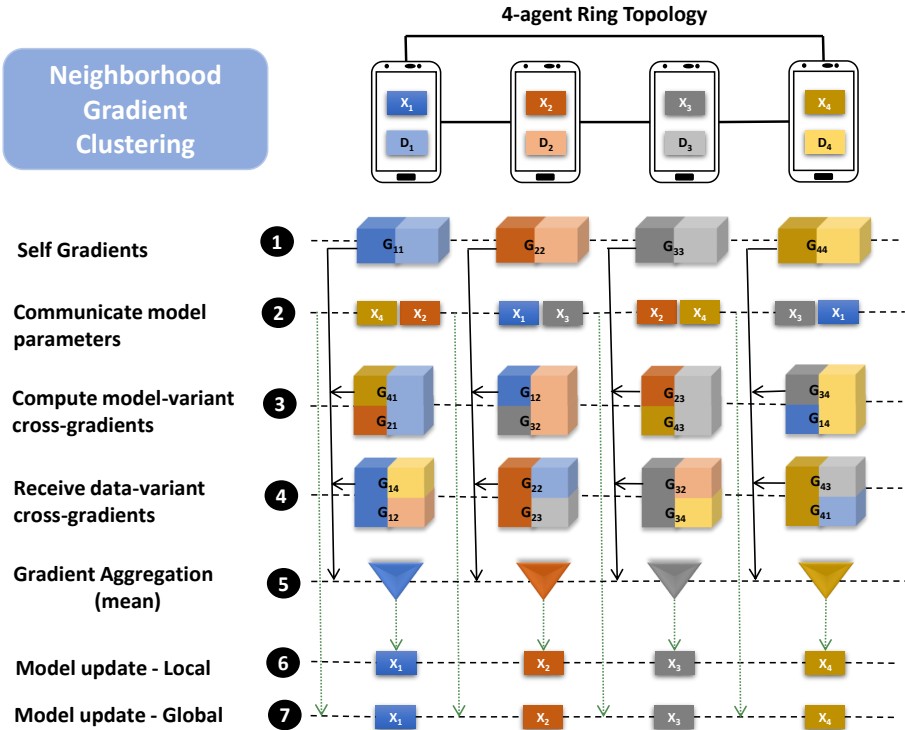

Figure 5: *NGM* algorithm overview. In the proposed algorithm, (1) each agent computes self-gradients of model parameters on its own data set; (2) each agent sends its model parameters to its neighbors; (3) each agent computes the model-variant cross-gradients of its neighbors' models on its own data set; (4) each agent receives the data-variant cross-gradients from its neighbors; (5) update the local gradient using the mean of self-gradients and cross-gradients ; (6) update the model parameter using local SGD step with updated gradients; (7) and update the model parameter using global gossip averaging step.

---

**Algorithm 2** Decentralized Peer-to-Peer Training (*D-PSGD* with momentum)

---

**Input:** Each agent $i \in [1, N]$ initializes model weights $x^i_{(0)}$, step size $\eta$, averaging rate $\gamma$, mixing matrix $W = [w_{ij}]_{i,j \in [1,N]}$, and $I_{ij}$ are elements of $N \times N$ identity matrix.

Each agent simultaneously implements the TRAIN( ) procedure
1. **procedure** TRAIN( )
2.     **for** k=0, 1, ..., $K - 1$ **do**
3.         $d^i_k \sim D^i$                                                     // sample data from training dataset.
4.         $g^i_k = \nabla_x f_i(d^i_k; x^k_i)$                                 // compute the local gradients
5.         $v^i_k = \beta v^i_{(k-1)} - \eta g^i_k$                             // momentum step
6.         $\widetilde{x}^i_k = x^i_k + v^i_k$                                  // update the model
7.         SENDRECEIVE($\widetilde{x}^i_k$)                                     // share model parameters with neighbors $N(i)$.
8.         $x^i_{(k+1)} = \widetilde{x}^i_k + \gamma \sum_{j \in \mathcal{N}_i}(w_{ij} - I_{ij})\widetilde{x}^j_k$   // gossip averaging step
9.     **end**
10. **return**

---

### A.4 CompNGM

The section presents the pseudocode for CompNGM (compressed version of $NGM_{dv}$) in Algorithm 3. The $NGM_{dv}$ algorithm at every iteration involves two rounds of communication with the neighbors: 1) communicate the model parameters, and 2) communicate the cross-gradients. This communication overhead can be a bottleneck in a resource-constrained environment. Hence we propose a compressed version of $NGM_{dv}$ using Error Feedback SGD (EF-SGD) (Karimireddy et al., 2019) to compress gradients. We compress the error-compensated self-gradients and cross-gradients from 32 bits (floating point precision of arithmetic operations) to 1 bit by using scaled signed gradients. The error between the compressed and non-compressed gradient of the current iteration is added as feedback to the gradients in the next iteration before compression.

Tables. 8, 9, 10 compare the proposed *CompNGM* with CompCGA and show that our method outperforms CompCGA. We also analyze the impact of compression on consensus error, validation loss, and validation accuracy with respect to communication bits in the top panel of Figure. 6. We present the consensus error, validation loss, and validation accuracy with respect to epochs in the bottom panel of Figure. 6. Note that epochs are proxies for communication rounds. In Figure. 6 each epoch contains 157 communication rounds. For the Figure. 6 experiment, we use a step-lr scheduler for the learning rate schedule whereas the experiments in Tables. 8, 9, 10 use the multistep-lr scheduler. We observe that both for iso-communication cost and iso-epochs, $NGM_{dv}$ performs better than *CompNGM* in terms of validation accuracy.

---

**Algorithm 3** Compressed Neighborhood Gradient Mean (*CompNGM*)

**Input:** Each agent $i \in [1, N]$ initializes model weights $x^i_{(0)}$, step size $\eta$, averaging rate $\gamma$, dimension of the gradient $d$, mixing matrix $W = [w_{ij}]_{i,j \in [1,N]}$, $NGM$ mixing weight $\alpha$, and $I_{ij}$ are elements of $N \times N$ identity matrix.

Each agent simultaneously implements the TRAIN( ) procedure
1. **procedure** TRAIN( )
2.     **for** k=0, 1, ..., $K-1$ **do**
3.         $d^i_k \sim D^i$         // sample data from training dataset
4.         $g^{ii}_k = \nabla_x f_i(d^i_k; x^i_k)$     // compute the local self-gradients
5.         $p^{ii}_k = g^{ii}_k + e^{ii}_k$     // error compensation for self-gradients
6.         $\delta^{ii}_k = (\|p^{ii}_k\|_1/d)sgn(p^{ii})$     // compress the compensated self-gradients
7.         $e^{ii}_{k+1} = p^{ii}_k - \delta^{ii}_k$     // update the error variable
8.         SENDRECEIVE($x^i_k$)     // share model parameters with neighbors $N(i)$
9.         **for** each neighbor $j \in \{N(i) - i\}$ **do**
10.             $g^{ji}_k = \nabla_x f_i(d^j_k; x^j_k)$     // compute neighbors' cross-gradients
11.             $p^{ji}_k = g^{ji}_k + e^{ji}_k$     // error compensation for cross-gradients
12.             $\delta^{ji}_k = (\|p^{ji}_k\|_1/d)sgn(p^{ji}_k)$     // compress the compensated cross-gradients
13.             $e^{ji}_{k+1} = p^{ji}_k - \delta^{ji}_k$     // update the error variable
14.             **if** $\alpha \neq 0$ **do**
15.                 SENDRECEIVE($\delta^{ji}_k$)     // share compressed cross-gradients between $i$ and $j$
16.             **end**
17.         **end**
18.         $\tilde{g}^i_k = (1 - \alpha) \sum_{j \in \mathcal{N}_i} w_{ji} \delta^{ji}_k + \alpha \sum_{j \in \mathcal{N}_i} w_{ij} \delta^{ij}_k$     // modify local gradients
19.         $v^i_k = \beta v^i_{(k-1)} - \eta \tilde{g}^i_k$     // momentum step
20.         $\tilde{x}^i_k = x^i_k + v^i_k$     // update the model
21.         $x^i_{(k+1)} = \tilde{x}^i_k + \gamma \sum_{j \in \mathcal{N}_i} (w_{ij} - I_{ij}) x^j_k$     // gossip averaging step
22.     **end**
23. **return**

---

Table 8: Average test accuracy comparisons for CIFAR-10 with non-IID data using various architectures and graph topologies. The results are averaged over three seeds where std is indicated.

| Method | Agents | 5layer CNN Ring | 5layer CNN Torus | VGG-11 Ring | ResNet-20 Ring |
|---|---|---|---|---|---|
| CompCGA | 5 | $82.00 \pm 0.25$ | - | $83.65 \pm 0.41$ | $86.73 \pm 0.34$ |
| | 10 | $71.41 \pm 0.94$ | $75.95 \pm 0.41$ | $73.96 \pm 0.31$ | $73.63 \pm 0.55$ |
| | 20 | $68.15 \pm 0.79$ | $71.71 \pm 0.54$ | $73.72 \pm 2.74$ | $66.34 \pm 0.98$ |
| *CompNGM* (ours) | 5 | $\mathbf{82.91 \pm 0.21}$ | - | $\mathbf{84.03 \pm 0.32}$ | $\mathbf{87.56 \pm 0.34}$ |
| | 10 | $\mathbf{74.36 \pm 0.42}$ | $\mathbf{77.82 \pm 0.20}$ | $\mathbf{77.02 \pm 0.14}$ | $\mathbf{78.50 \pm 0.98}$ |
| | 20 | $\mathbf{71.46 \pm 0.85}$ | $\mathbf{73.62 \pm 0.74}$ | $\mathbf{73.76 \pm 0.20}$ | $\mathbf{72.62 \pm 0.71}$ |

Table 9: Average test accuracy comparisons for various datasets with non-IID sampling trained over undirected ring topology. The results are averaged over three seeds where std is indicated.

| Method | Agents | Fashion MNIST (LeNet-5) | CIFAR-100 (ResNet-20) | Imagenette(32x32) (MobileNet-V2) |
|---|---|---|---|---|
| CompCGA | 5 | $90.45 \pm 0.34$ | $55.74 \pm 0.33$ | $72.76 \pm 0.44$ |
| | 10 | $81.62 \pm 0.37$ | $38.84 \pm 0.54$ | $59.92 \pm 0.72$ |
| *CompNGM* (ours) | 5 | $\mathbf{90.48 \pm 0.19}$ | $\mathbf{57.51 \pm 0.48}$ | $\mathbf{72.91 \pm 1.06}$ |
| | 10 | $\mathbf{83.38 \pm 0.39}$ | $\mathbf{43.07 \pm 0.32}$ | $\mathbf{61.91 \pm 2.10}$ |

Table 10: Average test accuracy comparisons for AGNews dataset (left side of the table) and full resolution $(224 \times 224)$ Imagenette dataset (right side of the table). The results are averaged over three seeds where std is indicated.

| Method | AGNews-BERT$_{mini}$ Agents = 4 | AGNews-BERT$_{mini}$ Agents = 8 | AGNews-DistilBERT$_{base}$ Agents = 4 | Imagenette $(224\times224)$-ResNet-18 Agents = 5 | Imagenette $(224\times224)$-ResNet-18 Agents = 5 |
|---|---|---|---|---|---|
| | Ring Topology | Ring Topology | Ring Topology | Ring Topology | Chain Topology |
| CompCGA | $91.05 \pm 0.29$ | $88.91 \pm 0.25$ | $\mathbf{93.54 \pm 0.03}$ | $84.65 \pm 0.57$ | $\mathbf{62.93 \pm 1.33}$ |
| *CompNGM* (ours) | $\mathbf{91.24 \pm 0.43}$ | $\mathbf{89.01 \pm 0.13}$ | $93.50 \pm 0.16$ | $\mathbf{85.44 \pm 0.10}$ | $62.64 \pm 0.85$ |

## A.5 Datasets

In this section, we give a brief description of the datasets used in our experiments. We use a diverse set of datasets each originating from a different distribution of images to show the generalizability of the proposed techniques.

**CIFAR-10:** CIFAR-10 (Krizhevsky et al., 2014) is an image classification dataset with 10 classes. The image samples are colored (3 input channels) and have a resolution of $32 \times 32$. There are $50,000$ training samples with 5000 samples per class and $10,000$ test samples with 1000 samples per class.

**CIFAR-100:** CIFAR-100 (Krizhevsky et al., 2014) is an image classification dataset with 100 classes. The image samples are colored (3 input channels) and have a resolution of $32 \times 32$. There are $50,000$ training samples with 500 samples per class and $10,000$ test samples with 100 samples per class. CIFAR-100 classification is a harder task compared to CIFAR-10 as it has 100 classes with very less samples per class to learn from.

**Fashion MNIST:** Fashion MNIST (Xiao et al., 2017) is an image classification dataset with 10 classes. The image samples are in greyscale (1 input channel) and have a resolution of $28 \times 28$. There are $60,000$ training samples with 6000 samples per class and $10,000$ test samples with 1000 samples per class.

**Imagenette:** Imagenette (Husain, 2018) is a 10-class subset of the ImageNet dataset. The image samples are colored (3 input channels) and have a resolution of $224 \times 224$. There are 9469 training samples with roughly 950 samples per class and 3925 test samples. We conduct our experiments on two different resolutions of

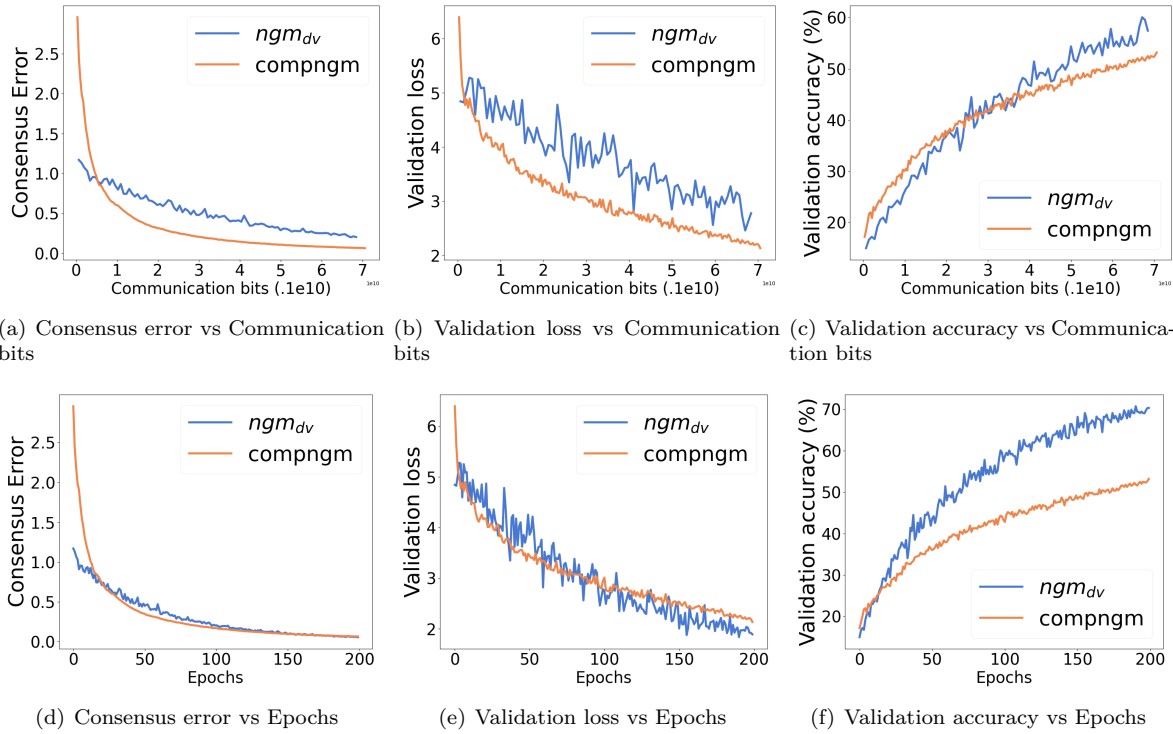

(a) Consensus error vs Communication bits

(b) Validation loss vs Communication bits

(c) Validation accuracy vs Communication bits

(d) Consensus error vs Epochs

(e) Validation loss vs Epochs

(f) Validation accuracy vs Epochs

Figure 6: Consensus error, validation loss, and validation accuracy against communication bits (top panel) and epochs (bottom panel) for training CIFAR-10 (non-IID) on ResNet-20 architecture over a ring topology with 10 agents.

the Imagenette dataset – a) a resized low resolution of $32 \times 32$ and, b) a full resolution of $224 \times 224$. The Imagenette experimental results reported in Table. 3 use the low-resolution images whereas experimental results in Table. 4 use the full resolution images.

**AGNews:** We use AGNews (Zhang et al., 2015) dataset for Natural Language Processing (NLP) task. This is a text classification dataset where the given text news is classified into 4 classes, namely "World", "Sport", "Business" and "Sci/Tech". The dataset has a total of 120000 and 7600 samples for training and testing respectively, which are equally distributed across each class.

### A.6 Network Architecture

We replace ReLU+BatchNorm layers of all the model architectures with EvoNorm-S0 (Liu et al., 2020a) as it was shown to be better suited for decentralized learning on non-IID data (Lin et al., 2021).

**5 layer CNN:** The 5-layer CNN consists of 4 convolutional with EvoNorm-S0 (Liu et al., 2020a) as an activation-normalization layer, 3 max-pooling layers, and one linear layer. In particular, it has 2 convolutional layers with 32 filters, a max pooling layer, then 2 more convolutional layers with 64 filters each followed by another max pooling layer and a dense layer with 512 units. It has a total of $76K$ trainable parameters.

**VGG-11:** We modify the standard VGG-11 (Simonyan & Zisserman, 2014) architecture by reducing the number of filters in each convolutional layer by $4\times$ and using only one dense layer with 128 units. Each convolutional layer is followed by EvoNorm-S0 as the activation-normalization layer and it has $0.58M$ trainable parameters.

**ResNet-20:** For ResNet-20 (He et al., 2016), we use the standard architecture with $0.27M$ trainable parameters except that BatchNorm+ReLU layers are replaced by EvoNorm-S0.

**LeNet-5:** For LeNet-5 (LeCun et al., 1998), we use the standard architecture with $61,706$ trainable parameters.

**MobileNet-V2:** We use the the standard MobileNet-V2 (Sandler et al., 2018) architecture used for CIFAR dataset with $2.3M$ parameters except that BatchNorm+ReLU layers are replaced by EvoNorm-S0.

**ResNet-18:** For ResNet-18 (He et al., 2016), we use the standard architecture with $11M$ trainable parameters except that BatchNorm+ReLU layers are replaced by EvoNorm-S0.

**BERT$_{\text{mini}}$:** For BERT$_{\text{mini}}$ (Devlin et al., 2018) we use the standard model from the paper. We restrict the sequence length of the model to 128. The model used in the work hence has $11.07M$ parameters.

**DistilBERT$_{\text{base}}$:** For DistilBERT$_{\text{base}}$ (Sanh et al., 2019) we use the standard model from the paper. We restrict the sequence length of the model to 128. The model used in the work hence has $66.67M$ parameters.

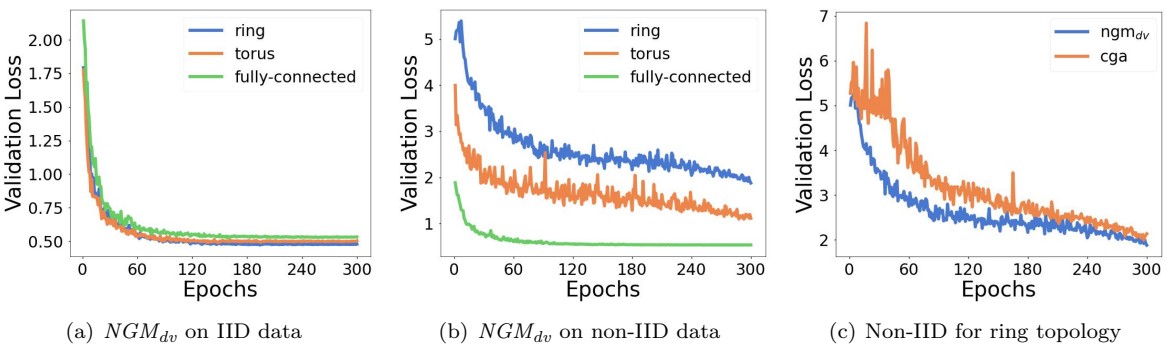

| (a) $NGM_{dv}$ on IID data | (b) $NGM_{dv}$ on non-IID data | (c) Non-IID for ring topology |

Figure 7: Validation loss for CIFAR-10 dataset trained on a 10 agents ring graph with 5-layer CNN.

### A.7 Analysis for 10 Agents

We show the convergence characteristics of the proposed $NGM_{dv}$ algorithm over IID and Non-IID data sampled from the CIFAR-10 dataset in Figure. 7(a), and 7(b) respectively. For Non-IID distribution, we observe that there is a slight difference in convergence rate (as expected) with a slower rate for sparser topology (undirected ring graph) compared to its denser counterpart (fully connected graph). Figure. 7(c) shows the comparison of the convergence characteristics of the $NGM_{dv}$ technique with the current state-of-the-art CGA algorithm. We observe that $NGM_{dv}$ has lower validation loss than CGA for the same decentralized setup indicating its superior performance over CGA. We also plot the model variance and data variance bias terms for both $NGM_{dv}$ and CGA techniques as shown in Figure. 8(a), and 8(b) respectively. We observe that both model variance and data variance bias for $NGM_{dv}$ are significantly lower than CGA.

### A.8 Resource Comparison

The communication cost, memory overhead, and compute overhead for various decentralized algorithms are shown in Table. 6. The D-PSGD, QG-DSGDm, and $NGM_{mv}$ algorithms require each agent to communicate model parameters of size $m_s$ with all the $n_i$ neighbors for the gossip averaging step and hence has a communication cost of $\mathcal{O}(n_i m_s)$. In the case of $NGM_{dv}$ and CGA, there is an additional communication round for sharing data-variant cross gradients apart from sharing model parameters for the gossip averaging step. So, both these techniques incur a communication cost of $\mathcal{O}(2n_i m_s)$ and therefore an overhead of $\mathcal{O}(n_i m_s)$ compared to D-PSGD. Momentum Tracking also needs additional communication to communicate the tracking variable. *CompNGM* compresses the additional round of communication involved with $NGM_{dv}$ from $b$ bits to 1 bit. This reduces the communication overhead from $\mathcal{O}(n_i m_s)$ to $\mathcal{O}(\frac{n_i m_s}{b})$.

CGA algorithm stores all the received data-variant cross-gradients in the form of a matrix for the quadratic projection step. Hence, CGA has a memory overhead of $\mathcal{O}(n_i m_s)$ compared to D-PSGD. $NGM_{dv}$ does not require any additional memory as it averages the received data-variant cross-gradients into the self-gradient

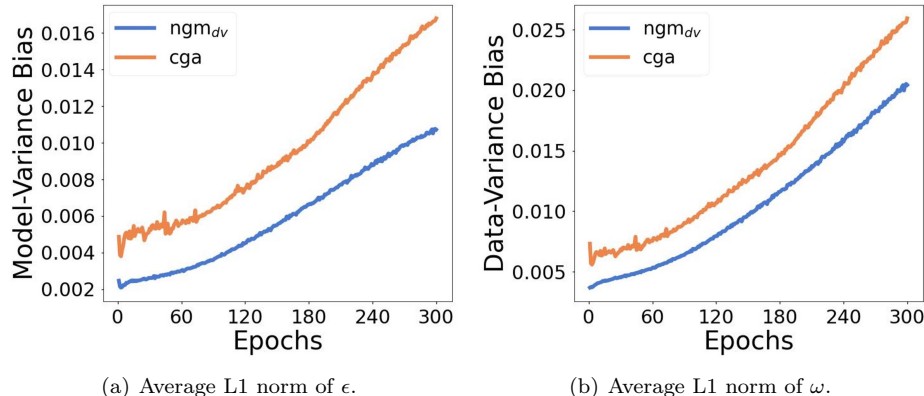

(a) Average L1 norm of $\epsilon$.

(b) Average L1 norm of $\omega$.

Figure 8: Average L1 norm of model variance bias and data variance bias for 10 agents trained on CIFAR-10 dataset with 5 layer CNN architecture over an undirected ring topology.

Table 11: Memory overheads for various methods trained on different model architectures with CIFAR-10 dataset over undirected ring topology with 2 neighbors per agent.

| Architecture | CGA (MB) | $NGM_{dv}$ (MB) | CompCGA (MB) | $CompNGM$ (MB) |
|---|---|---|---|---|
| 5 layer CNN | 0.58 | 0 | 0.58 | 0.60 |
| VGG-11 | 4.42 | 0 | 4.42 | 4.56 |
| ResNet-20 | 2.28 | 0 | 2.28 | 2.15 |

buffer. The compressed version of $NGM_{dv}$ requires an additional memory of $\mathcal{O}(n_i m_s)$ to store the error variables $e_{ji}$ (refer Algorith. 3). CompCGA also needs to store error variables along with the projection matrix of compressed gradients. Therefore, CompCGA has a memory overhead of $\mathcal{O}(n_i m_s + \frac{n_i m_s}{b})$. Note that memory overhead depends on the type of graph topology and model architecture but not on the size of the graph. The memory overhead for different model architectures trained on undirected ring topology is shown in Table. 11

The computation of the cross-gradients (in both CGA and NGM algorithms) requires $n_i$ forward and backward passes through the deep learning model at each agent. This is reflected as $\mathcal{O}(n_i B)$ in the compute overhead in Table. 6. We assume that the compute effort required for the backward pass is twice that of the forward pass. CGA algorithm involves quadratic programming projection step (Goldfarb & Idnani, 1983) to update the local gradients. Quadratic programming solver (quadprog) uses Goldfarb/Idnani dual algorithm. CGA uses quadratic programming to solve the following (Equation 47 -see Equation 5a in (Esfandiari et al., 2021)) optimization problem in an iterative manner:

$$\min_u \quad \frac{1}{2}u^T GG^T u + g^T G^T u$$
$$\text{s.t.} \quad u \geq 0 \tag{47}$$

where G is the matrix containing cross-gradients, g is the self-gradient, and the optimal gradient direction $g^*$ in terms of the optimal solution of the above equation $u^*$ is $g^* = G^T u^* + g$. The above optimization takes multiple iterations, resulting in compute and time complexity of polynomial(degree$\geq 2$) order. In contrast, NGM involves a simple averaging step that requires $O(n_i m_s)$ addition operations.

## A.9 Communication Cost

In this section we present the communication cost per agent in terms of Gigabytes of data transferred during the entire training process (refer Tables. 12, 13, 15, 14). The D-PSGD and $NGM_{mv}$ have the lowest

communication cost ($1\times$). We emphasize that $NGM_{mv}$ outperforms D-PSGD in decentralized learning over label-wise non-IID data for the same communication cost. $NGM_{dv}$ and CGA have $2\times$ communication overhead compared to D-PSGD where as $CompNGM$ and CompCGA have $1.03\times$ communication overhead compared to D-PSGD. The compressed versions of $NGM_{dv}$ and CGA compress the second round of cross-gradient communication to 1 bit. We assume the full-precision cross-gradients to be of 32-bit precision and hence the $CompNGM$ reduces the communication cost by $32\times$ compared to $NGM_{dv}$.

Table 12: Communication costs per agent in GBs for experiments in Table 2 and 8

| Method | Agents | 5layer CNN Ring | 5layer CNN Torus | VGG-11 Ring | ResNet-20 Ring |
|---|---|---|---|---|---|
| D-PSGD, | 5 | 17.75 | - | 270.64 | 127.19 |
| QG-DSGDm and | 10 | 8.92 | 13.38 | 135.86 | 63.84 |
| $NGM_{mv}$ | 20 | 4.50 | 6.65 | 68.48 | 32.18 |
| Momentum Tracking, | 5 | 35.48 | - | 541.05 | 254.27 |
| CGA and | 10 | 17.81 | 26.72 | 271.50 | 127.59 |
| $NGM_{dv}$ | 20 | 8.98 | 17.95 | 136.72 | 64.25 |
| CompCGA | 5 | 18.31 | - | 279.09 | 131.16 |
| and | 10 | 9.20 | 13.79 | 140.10 | 65.84 |
| $CompNGM$ | 20 | 4.64 | 9.28 | 70.61 | 33.18 |

Table 13: Communication costs per agent in GBs for experiments in Table 3 and 9

| Method | Agents | Fashion MNIST (LeNet-5) | CIFAR-100 (ResNet-20) | Imagenette (MobileNet-V2) |
|---|---|---|---|---|
| D-PSGD, QG-DSGDm and | 5 | 17.25 | 103.74 | 103.12 |
| $NGM_{mv}$ | 10 | 8.61 | 51.89 | 51.60 |
| Momentum Tracking, | 5 | 34.50 | 207.47 | 206.23 |
| CGA and $NGM_{dv}$ (ours) | 10 | 17.23 | 103.79 | 103.19 |
| CompCGA and | 5 | 17.79 | 106.98 | 106.34 |
| $CompNGM$ (ours) | 10 | 8.88 | 53.52 | 53.21 |

Table 14: Communication costs per agent in GBs for experiments in Table 4 and 10(right)

| Method | Ring topology | chain topology |
|---|---|---|
| D-PSGD, QG-DSGDm and and $NGM_{mv}$ | 501.98 | 401.59 |
| Momentum Tracking, CGA and $NGM_{dv}$ | 1003.96 | 803.17 |
| CompCGA and $CompNGM$ | 517.67 | 414.14 |

Table 15: Communication costs per agent in GBs for experiments in Table 4 and 10 (left)

| Method | BERT$_{mini}$ | | DistilBERT$_{base}$ |
|---|---|---|---|
| | Agents $= 4$ | Agents $= 8$ | Agents $= 4$ |
| D-PSGD, QG-DSGDm and $NGM_{mv}$ | 234.30 | 118.20 | 1410.39 |
| Momentum Tracking, CGA and $NGM_{dv}$ | 486.59 | 236.40 | 2820.77 |
| CompCGA and $CompNGM$ | 241.6 | 121.89 | 1454.46 |

## A.10 Hyper-parameters

This section presents the hyper-parameters for the experimental results presented in Sec. 5. All the experiments were run for three randomly chosen seeds. We decay the step size by 10x after 50% and 75% of the training for all the experiments except for figures. To generate the figures, the simulations are run for 300

Table 16: Hyper-parameters used for CIFAR-10 with non-IID data using various model architecture presented in Table 2 and  8

| | | 5 Layer CNN | | VGG-11 | ResNet-20 |
|---|---|---|---|---|---|
| Method | Agents | Ring | Torus | Ring | Torus |
| | (n) | $(\beta, \eta, \gamma)$ | $(\beta, \eta, \gamma)$ | $(\beta, \eta, \gamma)$ | $(\beta, \eta, \gamma)$ |
| D-PSGD | 5 | $(0.0, 0.1, 1.0)$ | $-$ | $(0.0, 0.01, 1.0)$ | $(0.0, 0.1, 1.0)$ |
| | 10 | $(0.0, 0.1, 1.0)$ | $(0.0, 0.1, 1.0)$ | $(0.0, 0.01, 1.0)$ | $(0.0, 0.1, 1.0)$ |
| | 20 | $(0.0, 0.1, 1.0)$ | $(0.0, 0.1, 1.0)$ | $(0.0, 0.01, 1.0)$ | $(0.0, 0.1, 1.0)$ |
| QG-DSGDm | 5 | $(0.9, 0.01, 1.0)$ | $-$ | $(0.9, 0.01, 1.0)$ | $(0.9, 0.1, 1.0)$ |
| | 10 | $(0.9, 0.01, 1.0)$ | $(0.9, 0.005, 0.1)$ | $(0.9, 0.01, 1.0)$ | $(0.9, 0.1, 1.0)$ |
| | 20 | $(0.9, 0.01, 1.0)$ | $(0.0, 0.005, 0.1)$ | $(0.9, 0.01, 1.0)$ | $(0.9, 0.1, 1.0)$ |
| $NGM_{mv}$ | 5 | $(0.0, 0.1, 1.0)$ | $-$ | $(0.9, 0.01, 1.0)$ | $(0.0, 0.1, 1.0)$ |
| | 10 | $(0.0, 0.1, 1.0)$ | $(0.0, 0.1, 1.0)$ | $(0.0, 0.01, 1.0)$ | $(0.0, 0.1, 1.0)$ |
| | 20 | $(0.0, 0.1, 1.0)$ | $(0.0, 0.1, 1.0)$ | $(0.0, 0.01, 1.0)$ | $(0.0, 0.1, 1.0)$ |
| Momentum Tracking | 5 | $(0.9, 0.01, 1.0)$ | $-$ | $(0.9, 0.01, 1.0)$ | $(0.9, 0.1, 1.0)$ |
| | 10 | $(0.9, 0.01, 1.0)$ | $(0.9, 0.01, 1.0)$ | $(0.9, 0.01, 1.0)$ | $(0.9, 0.1, 1.0)$ |
| | 20 | $(0.9, 0.01, 1.0)$ | $(0.0, 0.01, 1.0)$ | $(0.9, 0.01, 1.0)$ | $(0.9, 0.1, 1.0)$ |
| CGA and $NGM_{dv}$ | 5 | $(0.9, 0.01, 0.1)$ | $-$ | $(0.9, 0.1, 0.5)$ | $(0.9, 0.1, 1.0)$ |
| | 10 | $(0.9, 0.01, 0.5)$ | $(0.9, 0.01, 0.1)$ | $(0.9, 0.1, 0.5)$ | $(0.9, 0.1, 1.0)$ |
| | 20 | $(0.9, 0.01, 0.5)$ | $(0.9, 0.01, 0.1)$ | $(0.9, 0.1, 0.5)$ | $(0.9, 0.1, 1.0)$ |
| CompCGA and $CompNGM$ | 5 | $(0.9, 0.01, 0.1)$ | $-$ | $(0.9, 0.01, 0.1)$ | $(0.9, 0.01, 0.1)$ |
| | 10 | $(0.9, 0.01, 0.5)$ | $(0.9, 0.01, 0.1)$ | $(0.9, 0.01, 0.1)$ | $(0.9, 0.01, 0.1)$ |
| | 20 | $(0.9, 0.01, 0.5)$ | $(0.9, 0.01, 0.1)$ | $(0.9, 0.01, 0.1)$ | $(0.9, 0.01, 0.1)$ |

epochs with the StepLR scheduler where the learning rate is decayed by 0.981 after every epoch. We have used a momentum of 0.9 for all QG-DSGDm, Momentum Tracking, CGA, and $NGM_{dv}$ experiments. However, for D-PSGD and $NGM_{mv}$, the momentum is set to either 0.9 or 0.0 based on the validation accuracy. The averaging rate is always set to 1.0 for the Momentum Tracking algorithm.

**Hyper-parameters for CIFAR-10 :** All the experiments that involve 5-layer (Table. 2) have stopping criteria set to 100 epochs. We decay the step size by $10\times$ in multiple steps at $50^{th}$ and $75^{th}$ epoch. All the experiments for the CIFAR-10 dataset trained on VGG-11 and ResNet-20 (Table. 2) have stopping criteria set to 200 epochs. We decay the step size by $10\times$ in multiple steps at $100^{th}$ and $150^{th}$ epoch. Table 16 presents the $\beta$, $\eta$, and $\gamma$ corresponding to the momentum, step size, and gossip averaging rate.

**Hyper-parameters used for Table. 3:** All the experiments in Table. 3 have stopping criteria set to 100 epochs. We decay the step size by $10\times$ in multiple steps at $50^{th}$ and $75^{th}$ epoch. Table 17 presents the $\beta$, $\eta$, and $\gamma$ corresponding to the momentum, step size, and gossip averaging rate. For all the experiments related to Fashion MNIST and Imagenette (low resolution of $(32 \times 32)$), we use a mini-batch size of 32 per agent. For all the experiments related to CIFAR-100, we use a mini-batch size of 20 per agent. For all the experiments, we use a mini-batch size of 32 per agent.

**Hyper-parameters used for Table. 4 (right):** All the experiments in Table. 4 (right) have stopping criteria set to 100 epochs. We decay the step size by $10\times$ at $50^{th}, 75^{th}$ epoch. Table 18 (right) presents the $\beta$, $\eta$, and $\gamma$ corresponding to the momentum, step size, and gossip averaging rate. For all the experiments, we use a mini-batch size of 32 per agent.

**Hyper-parameters used for Table. 4 (left):** All the experiments in Table. 4 (left) have stopping criteria set to 3 epochs. We decay the step size by $10\times$ at $2^{nd}$ epoch. Table 18 (left) presents the $\beta$, $\eta$, and $\gamma$ corresponding to the momentum, step size, and gossip averaging rate. For all the experiments, we use a mini-batch size of 32 per agent on the AGNews dataset.

**Hyper-parameters used for Table. 7:** All the experiments in Table. 7 have stopping criteria set to 60 epochs. The initial learning rate is set to 0.01 and is decayed by $10\times$ at $20^{th}, 40^{th}, 54^{th}$ epoch. For all the experiments, we use a mini-batch size of 64 per agent. The values used for the learning rate, momentum coefficient, and averaging rate are presented in Table. 19.

Table 17: Hyper-parameters used for Table. 3 and 9

| Method | Agents (n) | Fashion MNIST $(\beta, \eta, \gamma)$ | CIFAR-100 $(\beta, \eta, \gamma)$ | Imagenette $(\beta, \eta, \gamma)$ |
|---|---|---|---|---|
| D-PSGD | 5 | $(0.0, 0.01, 1.0)$ | $(0.0, 0.1, 1.0)$ | $(0.0, 0.1, 1.0)$ |
|  | 10 | $(0.0, 0.01, 1.0)$ | $(0.0, 0.1, 1.0)$ | $(0.0, 0.1, 1.0)$ |
| QG-DSGDm | 5 | $(0.9, 0.01, 1.0)$ | $(0.9, 0.1, 1.0)$ | $(0.9, 0.01, 1.0)$ |
|  | 10 | $(0.9, 0.01, 1.0)$ | $(0.9, 0.1, 1.0)$ | $(0.9, 0.01, 1.0)$ |
| $NGM_{mv}$ | 5 | $(0.9, 0.01, 1.0)$ | $(0.0, 0.1, 1.0)$ | $(0.0, 0.1, 1.0)$ |
|  | 10 | $(0.9, 0.01, 1.0)$ | $(0.9, 0.1, 1.0)$ | $(0.0, 0.1, 1.0)$ |
| Momentum Tracking | 5 | $(0.9, 0.01, 1.0)$ | $(0.9, 0.1, 1.0)$ | $(0.9, 0.01, 1.0)$ |
|  | 10 | $(0.9, 0.01, 1.0)$ | $(0.9, 0.1, 1.0)$ | $(0.9, 0.01, 1.0)$ |
| CGA and $NGM_{dv}$ | 5 | $(0.9, 0.01, 1.0)$ | $(0.9, 0.1, 1.0)$ | $(0.0, 0.01, 0.5)$ |
|  | 10 | $(0.9, 0.01, 1.0)$ | $(0.9, 0.1, 1.0)$ | $(0.0, 0.01, 0.5)$ |
| CompCGA and $CompNGM$ | 5 | $(0.9, 0.01, 0.1)$ | $(0.9, 0.01, 0.1)$ | $(0.0, 0.01, 0.1)$ |
|  | 10 | $(0.9, 0.01, 0.1)$ | $(0.9, 0.01, 0.1)$ | $(0.0, 0.01, 0.5)$ |

Table 18: Hyper-parameters used for Table. 4 and 10

| Method | BERT$_{mini}$ Agents = 4 $(\beta, \eta, \gamma)$ | Agents = 8 $(\beta, \eta, \gamma)$ | DistilBERT$_{base}$ Agents = 4 $(\beta, \eta, \gamma)$ | Imagenette (ResNet-18) Ring topology $(\beta, \eta, \gamma)$ | Chain topology $(\beta, \eta, \gamma)$ |
|---|---|---|---|---|---|
| D-PSGD | $(0.0, 0.01, 1.0)$ | $(0.0, 0.01, 1.0)$ | $(0.0, 0.01, 1.0)$ | $(0.0, 0.01, 1.0)$ | $(0.0, 0.01, 1.0)$ |
| QG-DSGDm | $(0.9, 0.01, 1.0)$ | $(0.9, 0.01, 1.0)$ | $(0.9, 0.01, 1.0)$ | $(0.9, 0.01, 1.0)$ | $(0.9, 0.01, 1.0)$ |
| $NGM_{mv}$ (ours) | $(0.0, 0.01, 1.0)$ | $(0.0, 0.01, 1.0)$ | $(0.9, 0.01, 1.0)$ | $(0.9, 0.01, 1.0)$ | $(0.0, 0.01, 1.0)$ |
| Momentum Tracking | $(0.9, 0.005, 1.0)$ | $(0.9, 0.01, 1.0)$ | $(0.9, 0.01, 1.0)$ | $(0.9, 0.01, 1.0)$ | $(0.9, 0.01, 1.0)$ |
| CGA | $(0.9, 0.01, 0.5)$ | $(0.9, 0.01, 0.5)$ | $(0.9, 0.01, 0.5)$ | $(0.9, 0.01, 0.5)$ | $(0.9, 0.01, 0.1)$ |
| $NGM_{dv}$ | $(0.9, 0.01, 0.5)$ | $(0.9, 0.01, 0.5)$ | $(0.9, 0.01, 0.5)$ | $(0.9, 0.01, 0.5)$ | $(0.9, 0.01, 0.1)$ |
| CompCGA | $(0.9, 0.01, 0.5)$ | $(0.9, 0.01, 0.5)$ | $(0.9, 0.01, 0.5)$ | $(0.9, 0.01, 0.1)$ | $(0.9, 0.01, 0.1)$ |
| $CompNGM$ | $(0.9, 0.01, 0.5)$ | $(0.9, 0.01, 0.5)$ | $(0.9, 0.01, 0.5)$ | $(0.9, 0.01, 0.1)$ | $(0.9, 0.01, 0.1)$ |

Table 19: hyperparameters for Table. 7.

| Trained for 60 epochs | D-PSGD | QG-DSGDm | $NGM_{mv}$ | $NGM_{mv}$ +LM | $NGM_{mv}$ + QGM | $NGM_{dv}$ + LM | $NGM_{dv}$ + QGM |
|---|---|---|---|---|---|---|---|
| Learning rate ($\eta$) | 0.01 | 0.01 | 0.01 | 0.01 | 0.01 | 0.01 | 0.01 |
| Momentum coefficient ($\beta$) | 0.0 | 0.9 | 0.0 | 0.9 | 0.9 | 0.9 | 0.9 |
| Averaging rate ($\gamma$) | 0.5 | 1.0 | 0.5 | 1.0 | 1.0 | 1.0 | 1.0 |

