# OpenReview forum: "Neighborhood Gradient Mean: An Efficient Decentralized Learning Method for Non-IID Data"
_TMLR — Accepted by TMLR_

### Review · Reviewer_FMV6 · 2023-08-29

**Summary Of Contributions:**

This work studies decentralized optimization in the presence of non-iid data between the users. Following the lines of Esfandiari et al. (2021), it proposes a much more efficient way to aggregate the gradients of the neighbors (crossed gradients). While Esfandiari et al. (2021) aggregate through a hard to compute quadratic program, the algorithm presented in this work simply averages with the neighbors' gradients. From this idea, three different algorithms are proposed:
 - $NGC_{dv}$, for which the crossed gradients correspond to the gradient, with the local model and neighbors' data ($\nabla f_j(x_i)$ where $j$ is the neighbor and $i$ is the agent)
 - $NGC_{mv}$, for which the crossed gradients correspond to the gradient, with neighbors' model and local data ($\nabla f_i(x_j)$)
 - $CompNGC$ which is a corresponding version of $NGC_{dv}$ with compression of the sent information

The authors then yield a convergence rate of the optimization scheme, when following the algorithm $NGC_{dv}$. Finally, they compare the three proposed algorithms in an extensive numerical study with standard decentralized optimization algorithms.

**Audience:**

Yes

**Broader Impact Concerns:**

See my remark on the mention of privacy in the abstract

**Claims And Evidence:**

Yes

**Requested Changes:**

As explained above, I would like a clearer Theorem 1, as well as further explanations on it and Corollary 1. I would also like additional information in Table 1 (following all my remarks above).

Here are some last remarks:
 - in the abstract "without breaking the privacy constraints of the decentralized setting". I would remove the word privacy here, as such phrasing is sometimes used (by non-experts) to claim that decentralized optim implies privacy, while the leakage of gradients and model parameters are clearly not guarantee of privacy
 - "the word clustering is used as a synonym for grouping": why not directly use grouping then?
 - At the beginning of Section 3.1 "The flow of the neighborhood". *Flow* is inappropriate here, as it also has a mathematical, different meaning in optimization
 - why using the star for scalar multiplication all along the paper?
 - you use cross gradients as either $g^{ij}$ or $g^{ji}$. Have you considered using the $g^{jj}$ terms instead? I believe this is a known algorithm
 - I do not understand the compression scheme in Algorithm 3. In particular, when reading line 6, I do not get why $\delta$ could be sent as a single bit. It instead seems to be a (non binary) vector in $\mathbb{R}^d$.

**Strengths And Weaknesses:**

Strengths
===

This work nicely builds upon the work and setting of Esfandiari et al. (2021). It yields similar guarantees as the work of Esfandiari et al. (2021), using a similar set of assumptions, while requiring a much smaller computation time.
The overall claims seem sound.
Lastly, the experimental setup is quite extensive, illustrating the interest of $NGC$ algorithms on standard datasets.

Weaknesses
===
First, it is unclear in several parts of the paper whether it is about the optimization aspect of $NGC$, or its generalization aspect. In particular, the theoretical part (Section 4) seems to solely focus on the optimization aspect (see for example Table 1); while the experiments (Section 5) focus on the validation loss. The gap between these two aspects does not seem to be mentioned once in the paper, and should be explored more vastly in my opinion. I indeed believe that the reason for the good empirical performance of $NGC$ might not be because it quickly converges towards some stable point, but rather because this convergence point has some nice property.

Apart from that, my other concerns are not only specific to this work but shared with Esfandiari et al. (2021) and concern the theoretical part. First, I find Theorem 1 poorly stated and hard to fully understand. I think it would require a rewriting and further discussion on its implication:
 - it seems that $C_1$ can be negative (or zero). Is there a condition on $\beta$ to avoid this? (no condition on $\beta$ is actually stated in the theorem)
 - it is not totally clear whether the constants $C_2,C_3,C_4,C_5$ become small when $\eta$ is small (which I believe is the regime of interest). Indeed, the division by $C_1$ makes it unclear. Actually, I think more constants should be put into the variables $C_1,C_2,C_3,C_4,C_5$ to make the equation (11) clearer.

From Theorem 1, the authors deduce Corollary 1, which I also find poorly stated. Is $C$ in Corollary 1 a universal constant or does it depend in some of the problem constants? In particular, the spectral gap $1-\sqrt{\rho}$ is often a quantity of interest in the convergence rate of decentralized optimization schemes. Does $C$ depend on this quantity?
Also, where does the assumption $\zeta^2=\mathcal{O}(1/\sqrt{K})$ comes from? It seems very arbitrary to me, just because it yields some nice rate. Actually, I believe such an assumption is not verified in practice: such a small $\zeta$ should actually correspond to (nearly) iid data between the users.

Similarly, the comparison in Table 1 is unclear: I believe some important quantities such as $\rho, \zeta$ are hidden here.

---

> ### Author Response · Authors · 2023-08-30
> **Reply to Reviewer FMV6**
>
> We firstly thank the reviewer for the time and feedback.
> We answer the questions raised in the weakness section here:
> 1. The goal of the theory section was to show that the proposed algorithm has similar convergence rates as the other decentralized learning algorithms. We agree that the improvements of NGC come from the gradient mixing which makes the local gradients more similar to the global/averaged gradients. The decentralized convergence rate of NGC is the same as CGA however NGC estimates better gradients than CGA. We will clearly describe this in the analysis and discussion section. We plan to add a plot on consensus error in the analysis section that could clarify this further.
> 2. Yes, $C_1>0$ implies the following condition: $1-\frac{L \eta}{2}(\sqrt{1+\frac{4}{L \eta}}-1)  < \beta < 1$. We can simplify this further using Taylor series expansion as  $\frac{1}{L \eta} < \beta < 1$
> 3. We have simplified Theorem 1 as follows and we will add this to the rebutted version of the paper
> $\frac{1}{K}\sum_{k=0}^{K-1}  \mathbb{E}[||\nabla  \mathcal{F}(\bar{x}_k)||^2] \leq \frac{1}{C_1K}(\mathcal{F}(\bar{x}^0)-\mathcal{F}^*) +
>         C_2(\frac{\sigma^2}{N}+\zeta^2)+ C_3 \frac{(\sigma^2+\zeta^2)}{(1-\sqrt{\rho})^2}$
> where $C_1 = \frac{1}{2} (\frac{\eta}{1-\beta}-\frac{(1-\beta)}{\beta L})$, $C_{2} = \frac{20 \eta^2 L^2 \beta (\eta^2+\beta)}{(1-\beta)^2(\eta \beta L - (1-\beta)^2)}$, and $C_{3}=\frac{14 \eta^3 L^3 \beta}{(1-\beta)^2(\eta \beta L - (1-\beta)^2)}$.
> Here we can clearly observe that $C_2, C_3$ are small when $\eta$ is small.
>
> 4. The constant $C$ in Corollary 1 depends on three quantities $(1-\sqrt{\rho})$, $\sigma^2$ and $\zeta^2$
> 5. We improve the results presented in Table 1 as follows
>
> |Method | Rate | Assumption | Simplified Rate |
> |------------|----------------------|:--------------:|------------------------|
> |D-PSGD | $\mathcal{O}\Big(\frac{1+\sigma^2}{\sqrt{NK}} + \frac{N^2 \sigma^2 }{K (1-\rho)}+ \frac{N^2 \zeta^2}{K (1-\sqrt{\rho})^2} \Big)$ |  -|$\mathcal{O}(\frac{1}{\sqrt{NK}} + \frac{1}{K} )$|
> |CGA | $\mathcal{O}\Big(\frac{1+\sigma^2}{\sqrt{NK}} +\frac{N(\zeta^2+\sigma^2+\epsilon^2)}{K(1-\sqrt{\rho})^2}+(\frac{\sqrt{K}}{\sqrt{N}} + \frac{\sqrt{N}}{\sqrt{K}}) \epsilon^2 \Big)$ | $\epsilon^2 = \mathcal{O}(\frac{1}{K})$ | $\mathcal{O}(\frac{1}{\sqrt{NK}} + \frac{1}{K} + \frac{1}{K^{1.5}} + \frac{1}{K^{2}})$ |
> |$NGC_{dv}$ | $\mathcal{O}\Big(\frac{1+\sigma^2}{\sqrt{NK}} + \frac{N(\sigma^2 + \zeta^2)}{K (1-\sqrt{\rho})^2} + \frac{\sqrt{N}}{\sqrt{K}} \zeta^2\Big)$| $\zeta^2 = \mathcal{O}(\frac{1}{\sqrt{K}})$ | $\mathcal{O}(\frac{1}{\sqrt{NK}} + \frac{1}{K} + \frac{1}{K^{1.5}})$|
> |$NGC_{dv}$ | $\mathcal{O}\Big(\frac{1+\sigma^2}{\sqrt{NK}} + \frac{N(\sigma^2 + \zeta^2)}{K (1-\sqrt{\rho})^2} + \frac{\sqrt{N}}{\sqrt{K}} \zeta^2\Big)$| $\zeta^2 = \mathcal{O}(\frac{1}{(1-\sqrt{\rho})N})$ | $\mathcal{O}(\frac{1}{\sqrt{NK}} + \frac{1}{K} )$|
> |$NGC_{dv}$ | $\mathcal{O}\Big(\frac{1+\sigma^2}{\sqrt{NK}} + \frac{N(\sigma^2 + \zeta^2)}{K (1-\sqrt{\rho})^2} + \frac{\sqrt{N}}{\sqrt{K}} \zeta^2\Big)$| - | $\mathcal{O}(\frac{\sqrt{N}}{\sqrt{K}}+\frac{1}{\sqrt{NK}} + \frac{1}{K})$|
>
> We agree that $\zeta^2$ in the order of $\frac{1}{\sqrt{K}}$ results in $\zeta^2$ being small. Hence, we update the corollary with a more explained bound on $\zeta^2$ which is $\mathcal{O}(\frac{1}{(1-\sqrt{\rho})N})$ where we assume that the graphs with higher connectivity (lower spectral gap) can tolerate more non-IIDness. We also provide results when there is no assumption on $\zeta^2$.
>
>
> We answer the questions raised in "requested changes" here
> 1. We agree that there are no explicit privacy claims in our paper and we will remove that phrase from the abstract.
> 2. We will update the paper name to "Neighborhood Gradient Mean"
> 3. "The flow of the neighborhood" will be replaced with "The pseudo-code for the neighborhood"
> 4. We apologize for the confusion, we will remove the unnecessary * operator symbols.
> 5. $g_{jj}$ term is already considered in the gossip averaging step and doesn't need to be explicitly added. In particular, the gossip averaging step averages the neighborhood models using $x_i^{t+1} = \sum_j w_{ij} x_j^t$. Each of $x_j^t$ uses $g_{jj}$ for their local update, and therefore $g_{jj}$ is mixed with $x_i$ via the gossip averaging step.
> 6. $\delta^{ji}=(||p^{ji}||_1/d)sgn(p^{ji})$. Here, we can split $\delta$ into two components i.e., $\(||p^{ji}||_1/d)$ and $sgn(p^{ji})$. Now we can send these components separately. $\(||p^{ji}||_1/d)$ is a scalar value that requires 32 bits and $sgn(p^{ji})$ is a binary vector of the size $d$ that requires $d$ bits.
>
> We will update the paper with all these changes and refine the theory section in the rebuttal version

---

### Review · Reviewer_Xvci · 2023-09-12

**Summary Of Contributions:**

The paper introduces and evaluates several novel decentralized learning algorithms, including NGCmv, NGCdv, and CompNGC, specifically tailored for a non-iid data distribution setting.
Under the standard assumptions for decentralized optimization (outlined in Assumptions 1-3), a convergence rate is provided for the smooth non-convex regime, expressed as:

$$\begin{aligned} \frac{1}{K} \sum_{k=0}^{K-1} \mathbb{E}\left[\left\|\nabla \mathcal{F}\left(\bar{x}_k\right)\right\|^2\right] \leq & \frac{1}{C_1 K}\left(\mathcal{F}\left(\bar{x}^0\right)-\mathcal{F}^*\right)+\left(10 C_2+10 C_3 \frac{\eta^2 \beta^2}{(1-\beta)^4}+4 C_4\right)\left(\frac{\sigma^2}{N}+\zeta^2\right)+ \\ & C_5 \frac{14 \eta^2\left(\sigma^2+\zeta^2\right)}{(1-\beta)^2(1-\sqrt{\rho})^2},\end{aligned}$$

which, given specific choices for parameters $\eta$ and $\zeta^2$, can be simplified to:

$\frac{1}{K} \sum_{k=0}^{K-1} \mathbb{E}\left[\left\|\nabla \mathcal{F}\left(\overline{\mathbf{x}}_k\right)\right\|^2\right] \leq C\left(\frac{1}{\sqrt{N K}}+\frac{1}{K}+\frac{1}{K^{1.5}}\right)$

**Assumption 1 - Lipschitz Gradients:** Each function $f_i(x)$ is L-smooth.

**Assumption 2 - Bounded Variance:** The variance of the stochastic gradients is assumed to be bounded.
$$
\begin{gathered}
\mathbb{E}_{d \sim D_i}\left\|\nabla F_i(x ; d)-\nabla f_i(x)\right\|^2 \leq \sigma^2 \forall i \in[1, N] \\
\left\|\nabla f_i(x)-\nabla \mathcal{F}(x)\right\|^2 \leq \zeta^2 \forall i \in[1, N]
\end{gathered}
$$

**Assumption 3 - Doubly Stochastic Mixing Matrix:** The mixing matrix $W$ is a real doubly stochastic matrix with $\lambda_1(W)=1$ and

$$
\max ( |\lambda_2(W)| , |\lambda_N(W)| ) \leq \sqrt{\rho}<1
$$


The paper's primary contribution is the NGC mechanism (an acronym for Neighborhood Gradient Clustering). NGC categorizes gradients at each decentralized agent into three distinct classes:

1. **Self-gradients (local gradients)**: These are the derivatives computed locally at each agent based on its own dataset and model parameters.

2. **Model-variant cross-gradients**: These gradients are calculated by taking the derivatives of the model parameters received from neighboring agents with respect to the agent's local dataset. This computation is done after the agent receives model parameters from its neighbors, a step common in gossip-based decentralized algorithms.

3. **Data-variant cross-gradients**: These are computed by determining the derivatives of an agent's local model with respect to its neighbors' datasets. Getting these gradients requires an additional round of communication.

The NGC method then clusters these gradients into two groups:

a. **Model-variant cluster:** This merges the self-gradients and model-variant cross-gradients.

b. **Data-variant cluster:** This group integrates the self-gradients and data-variant cross-gradients.

The ultimate aim is to to account for the high variation of local gradients across neighboring agents due to the non-IID data distribution. In NGCmv, the local gradients are substituted with the averages from the model-variant cluster. In NGdv, they are replaced with the averages from the data-variant cluster.

For experimental validation, the proposed methods (NGCmv, NGCdv, and CompNGC) are compared against established algorithms such as D-PSGD [1], CGA, and CompCGA [2] across a range of machine learning tasks.

**References:**

[1] Xiangru Lian, et al. _Can decentralized algorithms outperform centralized algorithms? A case study for decentralized parallel stochastic gradient descent_. Advances in Neural Information Processing Systems, 30, 2017.

[2] Yasaman Esfandiari, et al. _Cross-gradient aggregation for decentralized learning from non-iid data_. In International Conference on Machine Learning, pp. 3036–3046. PMLR, 2021.

**Audience:**

No

**Broader Impact Concerns:**

This work is primarily theoretical. Hence, this work has no identifiable potential for negative societal impact.

**Claims And Evidence:**

Yes

**Requested Changes:**

1) Consider enhancing the literature review (according to the points listed in the Weaknesses section).

2) Add the missing proofs.

3) Conduct experimental comparisons with recent methods. One might look at the experimental setups from [11] and [12] for baseline methods.

**Strengths And Weaknesses:**

**Strengths:**
1) The main paper is clearly written, with its principal claims effectively outlined, making the text easy to follow;
2) The literature review includes pertinent papers related to the topic;
3) The authors propose novel algorithms suitable for non-convex objectives, with provable convergence rates;
4) The authors conducted a comprehensive set of experiments on vision tasks (5-layer CNN, VGG-11, ResNet-20, LeNet-5, MobileNet-V2, ResNet-18 on CIFAR-10, CIFAR-100, Fashion MNIST, and Imagenette datasets) as well as language problems (BERT and DistilBERT on AGNews). Furthermore, they varied different topologies: Ring, Chain, and Torus. In most cases, the proposed methods improved over existing baselines.

**Weaknesses**:
1) While the paper indeed addresses quite relevant problems of data heterogeneity [3, 4] that are commonly found in the federated learning setting [5], it overlooks some recent advancements in the topic of decentralized optimization, both with and without communication compression. In particular, the authors overlooked a well-known approach called Gradient Tracking [6, 7], which provably removes the dependence on $\zeta^2$ [8,9,10]. Recently, gradient tracking was also combined with momentum [11].
2) Unlike methods involving gradient tracking, the rate of the proposed method (NGCdv) depends on the heterogeneity parameter. This means that, in the worst case, the proposed method is still impacted by the non-iid'ness of the data.
3) The authors presented proof only for the NGCdv algorithm in the manuscript, whereas the convergence rate for the other two methods (NGCmv and CompNGC) is absent. A general analysis for an arbitrary $\alpha \in [0,1]$ mixing constant is also missing. Could the authors please clarify this point?
4) The authors also introduced a compressed version of the NGC algorithm named CompNGC. However, they missed pertinent references to recent decentralized methods with compression [12-15].
5) Theorem 1 introduces an additional (and rather restrictive) assumption on the step size:
   $$1-\frac{6 \eta^2 L^2}{(1-\beta)^2(1-\sqrt{\rho})}-\frac{4 L \eta}{(1-\beta)^2} \geq 0.$$
Nevertheless, there's no discussion on its feasibility. Multiple studies on similar topics, such as [3,10,11], do not impose on this assumption.
6) In Corollary 1, it's assumed that data dissimilarity $\zeta = O(1/\sqrt{K})$, while $K$ can be considerably large, as it's required that $K \geq \frac{36 N L^2}{r^2}$. This setting aligns more with the homogeneous scenario, which seems to contradict the paper's primary objective. Furthermore, in real-world settings, $\zeta^2$ can be substantially large, and its value, which depends on the data, is beyond user-level control.
7) The authors claim to have compared their proposed methods to the existing state-of-the-art. However, the baselines they used for comparison are from 2017 [2] and 2021 [1], which are somewhat outdated.

**References:**

[3] Anastasia Koloskova, Nicolas Loizou, Sadra Boreiri, Martin Jaggi, and Sebastian Stich. A unified theory of decentralized SGD with changing topology and local updates. In International Conference on Machine Learning, 2020.

[4] Kevin Hsieh, Amar Phanishayee, Onur Mutlu, and Phillip Gibbons. The non-IID data quagmire of decen- tralized machine learning. In International Conference on Machine Learning, 2020.

[5] Wang, Jianyu, et al. "A field guide to federated optimization." _arXiv preprint arXiv:2107.06917_ (2021).

[6] Paolo Di Lorenzo and Gesualdo Scutari. NEXT: in-network nonconvex optimization. In IEEE Transactions on Signal and Information Processing over Networks, 2016.

[7] Angelia Nedić, Alexander Olshevsky, and Wei Shi. Achieving geometric convergence for distributed opti- mization over time-varying graphs. In SIAM Journal on Optimization, 2017.

[8] Hanlin Tang, Xiangru Lian, Ming Yan, Ce Zhang, and Ji Liu. d2: Decentralized training over decentralized data. In International Conference on Machine Learning, 2018b.

[9] Thijs Vogels, Lie He, Anastasia Koloskova, Sai Praneeth Karimireddy, Tao Lin, Sebastian U Stich, and Martin Jaggi. RelaySum for decentralized deep learning on heterogeneous data. In Advances in Neural Information Processing Systems, 2021.

[10] Anastasia Koloskova, Tao Lin, and Sebastian U Stich. An improved analysis of gradient  tracking for decen- tralized machine learning. In Advances in Neural Information Processing Systems, 2021.

[11] Takezawa, Yuki, Han Bao, Kenta Niwa, Ryoma Sato, and Makoto Yamada. "Momentum tracking: Momentum acceleration for decentralized deep learning on heterogeneous data." _arXiv preprint arXiv:2209.15505_ (2022).

[12] Haoyu Zhao, Boyue Li, Zhize Li, Peter Richtarik, and Yuejie Chi. Beer: Fast O(1/t) rate for decentralized nonconvex optimization with communication compression. In Advances in Neural Information Processing Systems, 2022.

[13] Yucheng Lu and Christopher De Sa. Moniqua: Modulo quantized communication in decentralized SGD. In International Conference on Machine Learning, 2020.

[14] Xiaorui Liu, Yao Li, Rongrong Wang, Jiliang Tang, and Ming Yan. Linear convergent decentralized opti- mization with compression. In International Conference on Learning Representations, 2021.

[15] Yuki Takezawa, Kenta Niwa, and Makoto Yamada. Communication compression for decentralized learning with operator splitting methods. In arXiv, 2022a.

---

> ### Author Response · Authors · 2023-09-22
> **Reply to Reviewer Xvci**
>
> We firstly thank the reviewer for the time and feedback. We answer the questions raised in the weakness section here:
>
> 1. We have added the reference to Gradient Tracking and Momentum Tracking Algorithms in the latest version of the paper. We have also included the experimental results on Momentum Tracking as a baseline for comparison.
> 2. Yes, the derived convergence rate for NGC depends on the heterogeneity parameter but it improves the performance on non-IID data empirically similar to QGM and CGA.
> 3. We presented the analysis of convergence rate for $NGC_{dv}$ i.e., for $\alpha=1$ case as the NGC gradients take a special structure in this case which improves the bound on the difference between the original and NGC gradient. For the other cases where $\alpha\neq 1$, the convergence analysis remains the same as CGA as mentioned in Sec 4. Further, it has been shown in the literature  [1,2] that communication compression with error feedback only affects the higher-order terms of the convergence rate. Hence, decentralized compression algorithms have linear speedup similar to D-PSGD. We leave the analysis to show that CompNGC has a similar convergence rate as NGC for future work.
> 4. The references mentioned by the reviewer are added to the latest version of the paper.
> 5. The discussion on step size and momentum coefficient is presented in Appendix A.2.3. A similar condition on step size can also be seen in the CGA algorithm's analysis
> 6. We improve the results presented in Table 1 as follows
>
>
>
> |Method | Rate | Assumption | Simplified Rate |
> |------------|----------------------|:--------------:|------------------------|
> |D-PSGD | $\mathcal{O}\Big(\frac{1+\sigma^2}{\sqrt{NK}} + \frac{N^2 \sigma^2 }{K (1-\rho)}+ \frac{N^2 \zeta^2}{K (1-\sqrt{\rho})^2} \Big)$ |  -|$\mathcal{O}(\frac{1}{\sqrt{NK}} + \frac{1}{K} )$|
> |CGA and $NGC_{mv}$   | $\mathcal{O}\Big(\frac{1+\sigma^2}{\sqrt{NK}} +\frac{N(\zeta^2+\sigma^2+\epsilon^2)}{K(1-\sqrt{\rho})^2}+(\frac{\sqrt{K}}{\sqrt{N}} + \frac{\sqrt{N}}{\sqrt{K}}) \epsilon^2 \Big)$ | $\epsilon^2 = \mathcal{O}(\frac{1}{K})$ | $\mathcal{O}(\frac{1}{\sqrt{NK}} + \frac{1}{K} + \frac{1}{K^{1.5}} + \frac{1}{K^{2}})$ |
> |$NGC_{dv}$ | $\mathcal{O}\Big(\frac{1+\sigma^2}{\sqrt{NK}} + \frac{N(\sigma^2 + \zeta^2)}{K (1-\sqrt{\rho})^2} + \frac{\sqrt{N}}{\sqrt{K}} \zeta^2\Big)$| $\zeta^2 = \mathcal{O}(\frac{1}{\sqrt{K}})$ | $\mathcal{O}(\frac{1}{\sqrt{NK}} + \frac{1}{K} + \frac{1}{K^{1.5}})$|
> |$NGC_{dv}$ | $\mathcal{O}\Big(\frac{1+\sigma^2}{\sqrt{NK}} + \frac{N(\sigma^2 + \zeta^2)}{K (1-\sqrt{\rho})^2} + \frac{\sqrt{N}}{\sqrt{K}} \zeta^2\Big)$| $\zeta^2 = \mathcal{O}(\frac{1}{(1-\sqrt{\rho})N})$ | $\mathcal{O}(\frac{1}{\sqrt{NK}} + \frac{1}{K} )$|
> |$NGC_{dv}$ | $\mathcal{O}\Big(\frac{1+\sigma^2}{\sqrt{NK}} + \frac{N(\sigma^2 + \zeta^2)}{K (1-\sqrt{\rho})^2} + \frac{\sqrt{N}}{\sqrt{K}} \zeta^2\Big)$| - | $\mathcal{O}(\frac{\sqrt{N}}{\sqrt{K}}+\frac{1}{\sqrt{NK}} + \frac{1}{K})$|
>
> We agree that $\zeta^2$ in the order of $\frac{1}{\sqrt{K}}$ results in $\zeta^2$ being small. Hence, we update the corollary with a more explained bound on $\zeta^2$ which is $\mathcal{O}(\frac{1}{(1-\sqrt{\rho})N})$ where we assume that the graphs with higher connectivity (lower spectral gap) can tolerate more non-IIDness. We also provide results when there is no assumption on $\zeta^2$.
>
> 7. As per the reviewer's suggestion, we have added the experimental results on QG-DSGDm and Momentum Tracking Algorithms to the list of comparison baselines.
>
> We answer the questions raised in "requested changes" here
>
> 1. We have improved the literature presented in the paper by adding references to the recent papers as suggested by the reviewer.
> 2. We have improved the theory section and updated Table 1. A discussion on step size and momentum coefficient is presented in the Appendix. We added a discussion on CompNGC convergence analysis in section 6.
> 3. We added experimental comparisons with recent methods such as QG-DGSDm [3] and Momentum Tracking [4] in the latest version of the paper. We also added experiments to show that QGM [3] can be used in synergy with NGC to improve the accuracy furthrt.
>
>  References
>
> [1] Koloskova, Anastasia, Sebastian Stich, and Martin Jaggi. "Decentralized stochastic optimization and gossip algorithms with compressed communication." International Conference on Machine Learning. PMLR, 2019.
>
> [2] Tang, Hanlin, et al. "$\texttt {DeepSqueeze} $: Decentralization Meets Error-Compensated Compression." arXiv preprint arXiv:1907.07346 (2019).
>
> [3] Lin, Tao, et al. "Quasi-global momentum: Accelerating decentralized deep learning on heterogeneous data." arXiv preprint arXiv:2102.04761 (2021).
>
> [4] Takezawa, Yuki, Han Bao, Kenta Niwa, Ryoma Sato, and Makoto Yamada. "Momentum tracking: Momentum acceleration for decentralized deep learning on heterogeneous data." arXiv preprint arXiv:2209.15505 (2022).

---

> > ### Comment · Reviewer_Xvci · 2023-09-27
> > **Further updates**
> >
> > Dear Authors,
> >
> > Thank you for the provided revision.
> > I appreciate your attention to my previous comments concerning the literature review and experimental setup. Thank you for making the necessary adjustments. However, I would like to point out that the paper [1], which I previously referenced, has not been incorporated as a practical baseline in your study.
> > I believe that this particular paper would serve as a significant competitor to CompNGM.
> > It is essentially an EF21-type method (a novel notion of error feedback appeared in [2]) designed for a decentralized setup.
> > It also has a built-in gradient tracking since its rate is independent of heterogeneity parameter $\zeta^2$.
> >
> > **Suggestions for the future revision:**
> >
> > 1) I kindly suggest considering the inclusion if the paper [1] as a baseline in your work to enhance the comparison and overall evaluation of CompNGM.
> >
> > 2) Also, in Tables 2 and 3, compressed and uncompressed methods are compared together.
> > However, it would also be interesting to highlight either the wall-clock time for the method to reach the presented accuracy or the number of bits communicated since methods with compression are usually more beneficial in terms of that metric than non-compressed ones.
> >
> > 3) Also, for papers presenting some communication-efficient guarantees (eg, [3]), it is a standard to show convergence behavior in terms of the number of bits communicated per worker. It would also be beneficial to highlight when comparing Comp-NGM versus other baselines.
> >
> > **References:**
> >
> > [1]  Haoyu Zhao, Boyue Li, Zhize Li, Peter Richtarik, and Yuejie Chi. Beer: Fast O(1/T) rate for decentralized nonconvex optimization with communication compression. In Advances in Neural Information Processing Systems, 2022.
> >
> > [2] Peter Richtárik, Igor Sokolov, and Ilyas Fatkhullin. "EF21: A new, simpler, theoretically better, and practically faster error feedback." Advances in Neural Information Processing Systems 34 (2021): 4384-4396.
> >
> > [3] Eduard Gorbunov, et al. "MARINA: Faster non-convex distributed learning with compression." International Conference on Machine Learning. PMLR, 2021.

---

> ### Author Response · Authors · 2023-09-29
> **Reply to further updates from Reviewer Xvci**
>
> We thank the reviewer for the feedback on our rebuttal. We reply to the points raised in "Suggestions for the future revision" here.
>
> 1. We believe that BEER [1] is not a required baseline for comparison for the following reasons: BEER adopts communication compression with the gradient tracking algorithm. It has been shown in the QGM [4] paper that the gradient tacking algorithms do not scale well for deep learning models on non-IID data and that QG-DSGDm performs better than gradient tracking. This would mean that gradient tracking with compression (BEER) will not outperform QG-DSGDm on non-IID data. Further, our experiments show that CompNGM performs better than Momentum Tracking by at least 2% in computer vision tasks (except for Fashion MNIST). Therefore, we can conclude that a compressed version of momentum tracking (or gradient tracking) will not outperform CompNGM. In addition, we would like to highlight that CompNGM is just one compressed version of NGM and one could potentially combine NGM with better compression methods. The reason behind our analysis is to show that we can trade-off accuracy for reducing communication costs.
> 2. We reorganized the paper according to this suggestion and added a separate section for communication compression results (sec 5.4 in the latest revision). Table. 5 highlights the accuracy drop caused due to the communication compression (comparing the trade-off between performance and communication cost).
> 3. Thanks for pointing this out. We updated Appendix A.4 based on this suggestion and added a new figure (Figure. 6)  showing the convergence behavior in terms of the number of bits communicated per worker.
>
> We have addressed all the concerns raised by the reviewer. Please let us know if you have any further suggestions or need any clarifications.
>
> Reference:
>
> [4] Lin, Tao, et al. "Quasi-global momentum: Accelerating decentralized deep learning on heterogeneous data." arXiv preprint arXiv:2102.04761 (2021).

---

### Review · Reviewer_Uzo9 · 2023-09-18

**Summary Of Contributions:**

This paper presents a novel decentralized learning algorithm that leverages neighborhood gradient clustering in a network to mitigate the issue of poor performance due to non-IID data distributions among different agents. The authors propose NGC in which the local gradients are modified using self- and cross-gradient information, which includes model-variant cross-gradients and data-variant cross-gradients. They also develop a compressed version of NGC, termed CompNGC to reduce the communication overhead. The authors present the theoretical analysis to show the convergence rate of the proposed algorithm. To validate NGC/CompNGC, the authors show thorough empirical results to demonstrate the outperforming capability over the baselines using different datasets and models, with different topologies.

**Audience:**

Yes

**Claims And Evidence:**

Yes

**Requested Changes:**

Please add some theoretical analysis or discussion for the CompNGC.

**Strengths And Weaknesses:**

Strengths:

The paper is well written and easy to follow. The presentation in this paper is clear, while it requires a little more technical details and clarifications on explaining the proposed algorithm. The theory looks good and empirical results are comprehensive and promising, which include different tasks, datasets, and models.

Weaknesses:

The novelty of this paper is marginal. It looks like the authors develop their methodologies on top of the paper proposing CGA. Comparing these two papers, the only theoretical difference is that the authors relax the assumption for the difference between the original and included gradient and present Lemma 1 to bound it. Other than that, the theoretical analysis looks pretty similar, in terms of convergence rate and proof techniques, regardless of different terminologies and coefficients. I don’t think this would add great values to the paper. Though we can observe the slight improvements brought by NGC for the accuracies, the significance of the proposed algorithm is not that much.

---

> ### Author Response · Authors · 2023-09-18
> **Reply to Reviewer Uzo9**
>
> We firstly thank the reviewer for the time and feedback. We answer the questions raised in the weakness section here:
> We would like to reiterate the contributions of our work -
> * We demonstrate that $NGC_{mv}$  improves the performance on non-IID data significantly (0.19-34.37% with an average of 11.96% across all the experiments) compared to D-PSGD. To the best of our knowledge, there is no work that utilizes the locally available model-variant cross-gradients to improve the decentralized learning performance on non-IID data with no communication overhead.
> * We agree that $NGC_{dv}$ has marginal improvements over CGA in some cases. However, it is important to note that $NGC$ is significantly more efficient than CGA in compute, memory, and latency (please refer to section 5.4). Further, Table. 6 shows that our algorithm scales for larger datasets like ImageNet unlike CGA (which is $20\times$ slower).
> * The main contribution of NGC is on how to utilize the cross-gradients in an efficient way. The gradient update of $NGC_{dv}$ has a nice structure, giving a better bound for the difference between the original and NGC gradients. Hence, Lemma 1 is the critical part of our proof and the remaining part follows from CGA.
>
> We answer the questions raised in "requested changes" here
>
> The convergence rate for CompNGC is also of the same order as NGC. It has been shown in several decentralized compression works in the literature [1, 2] that communication compression with error feedback only affects the higher order terms and hence will still have linear speedup similar to DPSGD. We will include this discussion in the section. 6.
>
> References:
>
> [1] Koloskova, Anastasia, et al. "Decentralized deep learning with arbitrary communication compression." arXiv preprint arXiv:1907.09356 (2019).
>
> [2] Tang, Hanlin, et al. "DeepSqueeze: Decentralization Meets Error-Compensated Compression." arXiv preprint arXiv:1907.07346 (2019).

---

### Author Response · Authors · 2023-09-22
**Updates made to the manuscript**

The following updates are made to the latest version of the paper as part of the rebuttal
1. The name of the paper has been changed to "Neighborhood Gradient Mean" as per the reviewer's request.
2. We have improved Theorem 1 by simplifying the coefficients. We also added the constraint on momentum coefficient $\beta$.
3. We have updated Table 1 to include $\sigma^2$, $\zeta^2$, and $\rho$ in the final convergence rate equation.
4. We have added two more baselines (QG-DSGDm and Momentum Tracking) to our experimental section. Tables 2,3 and 4 are updated accordingly. Note that for a fair comparison with QG-DSGDm, we updated the $NGM_{mv}$ results to include local momentum in the cases where local momentum helps.
5. We included additional figures (4a and 4b) showing the consensus error plot and the effect of combining QGM with the proposed method.
6. We updated Table 5 to include the new baselines.
7. We have updated the literature review to include the missing references.
8. We added a discussion on CompNGC convergence analysis in section 6.

We have addressed all the concerns raised by the reviewers and would be happy to clarify any further questions.

---

### Decision · Action_Editor_Gns6 · 2023-10-26

**Recommendation:** Accept with minor revision

**Comment:**

The reviewers felt that the advancements over Esfandiari et al. (2021) were somewhat incremental, but they were unique. As such, we determined that the submission meets the TMLR acceptance standards.

### Minor revision
- Please note that the references are not only hard to distinguish from the text (as in "D-PSGD Lian et al. (2017), QG-DSGDm
Lin et al. (2021), Momentum Tracking Takezawa et al. (2022), and CGA Esfandiari et al. (2021)"), but also do not follow the TMLR style file (Section 4.1). Please update accordingly for the final version.

- Please (if possible) simplify the conditions on the stepsize (Equations (1), (2)). One of the reviewers pointed this out in the internal discussion.

**Audience:**

Yes, the presented results offer valuable insights, especially from a practical standpoint, and are of interest to the TMLR audience.

**Claims And Evidence:**

This paper explores decentralized optimization when compute nodes have non-iid data. Drawing inspiration from (Esfandiari et al., 2021), this paper introduces a more streamlined method for combining the gradients from neighboring data points (referred to as crossed gradients).

The paper gives a convergence analysis of the proposed method and - after the revision - the reviewers agree that the assumptions and theorems are clearly stated and accurate.

The experimental study is adding further insights in the performance of the proposed method against the relevant baselines (although verified only for a relatively small number of nodes, up to 20).